# Hierarchical multi-timescale structural dynamics of the disordered N-terminal of p53

Dániel Szöllősi [1,11], Supriya Pratihar[2,6,11], Dwaipayan Mukhopadhyay[2], Ashok Kumar Rout[2,7], Mookyoung Han [2], G. Jithender Reddy [2,8], Niklas Ebersberger [2,9], Stefan Becker [2], Gábor Nagy [1], Sarah Rauscher [3,4,5], Donghan Lee[2,10], Reinhard Klement[1], Christian Griesinger [2] ✉ & Helmut Grubmüller [1] ✉

Most natively folded proteins exhibit a unique spatial structure, which undergoes functional motions ranging from picoseconds to seconds, governed by a hierarchically ordered, funnel-shaped free energy landscape. Intrinsically disordered proteins (IDPs) lack such a stable native structure, but undergo fast interconversions between many different structures. Accordingly, the underlying free energy landscape is assumed to be rather shallow and unstructured. However, although IDPs represent nearly one-third of the human proteome, their structural dynamics on timescales slower than nanoseconds remain largely elusive. Here we reveal the structural dynamics of the prototypical IDP p53-TAD, also known as the "guardian of the genome", by combining high-power relaxation dispersion nuclear magnetic resonance spectroscopy with large-scale molecular dynamics simulations. We found a complex hierarchy of structural dynamics on timescales covering over seven orders of magnitude, ranging from fast nanoseconds backbone reorientations, via sub-microsecond helix-formation dynamics involving many structural substates and transition times, to transient tertiary structure formation slower than 25 microseconds. These rich structural dynamics are unexpectedly similar to the timescale hierarchy of natively folded proteins, which may be key to the ability of p53-TAD – and possibly of other IDPs – to bind many different partners by folding into different structures.

Folded proteins exhibit complex structural dynamics across a wide range of timescales, covering picoseconds to milliseconds or even longer which often implements or modulates their biochemical function[1–4]. These dynamics are governed by a complex[2,5,6], funnel-shaped free energy landscape[7], characterised by a hierarchy of low to high energy barrier "tiers"[8]. Detailed knowledge of the distribution of these energy barriers, the resulting kinetics, and the underlying structural dynamics are therefore key to our understanding of how

[1]Max Planck Institute for Multidisciplinary Sciences, Department of Theoretical and Computational Biophysics, Göttingen, Germany. [2]Max Planck Institute for Multidisciplinary Sciences, Department of NMR-Based Structural Biology, Göttingen, Germany. [3]Department of Chemical and Physical Sciences, University of Toronto Mississauga, Mississauga, ON, Canada. [4]Department of Physics, University of Toronto, Toronto, ON, Canada. [5]Department of Chemistry, University of Toronto, Toronto, ON, Canada. [6]Present address: Columbia University, Department of Biochemistry and Molecular Biophysics, New York, NY, USA. [7]Present address: Institute of Chemistry and Metabolomics, University of Lübeck, Lübeck, Germany. [8]Present address: NMR Division, Department of Analytical & Structural Chemistry, CSIR-Indian Institute of Chemical Technology, Hyderabad, India. [9]Present address: Institute of Biochemistry, University of Lübeck, Lübeck, Germany. [10]Present address: Biopharmaceutical Research Center, Korea Basic Science Institute, Cheongju-Si, South Korea. [11]These authors contributed equally: Dániel Szöllősi, Supriya Pratihar. ✉e-mail: cigr@mpinat.mpg.de; hgrubmu@mpinat.mpg.de

proteins fold and how they perform their remarkably broad range of biochemical functions on the molecular level. At the opposite end of the structural continuum[9] lie intrinsically disordered peptides and proteins (IDPs) which, in contrast, lack such a stable native structure[10,11]. Rather, they rapidly interconvert between many different structures, with only transiently formed secondary structure elements[12] and long-range contacts[13]. Operating at the limits of Anfinsen's dogma[14], these fast reconfiguration dynamics[15,16] gave rise to the general notion that the underlying free energy landscape is rather shallow and unstructured[17,18] or weakly funnelled[19].

IDPs pose considerable experimental and computational challenges, however, and hence information on their kinetics and specifically their structural dynamics is still sparse. For example, single-molecule spectroscopy experiments revealed fast (ns) reconfiguration dynamics of several IDPs[16,20,21]. Combining start-stop perturbation experiments with fluorescence correlation spectroscopy measurements enabled probing timescales from picoseconds to seconds[15,20], yet without revealing atomistic details of the underlying structural dynamics. Kinetic information at fast (ps to ns) timescales is also provided by nuclear magnetic resonance (NMR) relaxation measurements[22,23], whereas NMR relaxation dispersion (RD) measurements typically probe kinetics at several tens of µs or slower[24]. High-speed atomic force microscopy probes dynamics on the ms timescale with near residue level spatial resolution[25]. Atomistic molecular dynamics (MD) simulations provide more direct structural insights, but typically only quantify sub-µs structural dynamics[26–29], although extensions to µs timescale have been achieved using Markov state models, e.g., for Aβ42[30]. Deep learning based techniques also aim to predict structural ensembles at reduced computational cost, yet so far without interconversion dynamics or timescale information[31,32]. Thus, while combined fluorescent techniques, in particular Förster resonance energy transfer (FRET) and nanosecond fluorescence correlation spectroscopy (nsFCS) access timescales between 100 ns and 10 µs[20,33], directly linking the kinetics of IDPs to their structural dynamics at atomistic detail, particularly at these timescales, has not yet been possible.

Here we combined high-power NMR RD measurements and milliseconds atomistic MD simulations to gain access to these 100 ns to 10 µs kinetics and structural dynamics of a prototypic IDP, the intrinsically disordered N-terminal transactivation domain of the tumour protein p53 (p53-TAD, Fig. 1A). Also known as the "guardian of the genome", p53 regulates the cellular response to genomic damages and thus prevents cancer formation[34]. As one of the most important signalling hubs, p53-TAD adopts different structures upon specific binding to a stunningly large number of proteins[35,36]. X-ray structures of the p53/MDM2 complex[35], NMR measurements[12], and MD simulations[26] suggest that the TAD in solution forms transient helical structures ("helix 1" and "helix 2" in Fig. 1A)[37]. These helices cooperatively engage in binding, can adopt well-folded conformations upon interaction with binding partners[23,35], and affect the autoinhibition of p53[38,39]. Further, their stability in the unbound state modulates the lifetime of the bound complex[40,41]. The structural dynamics of this prototypical IDP might therefore regulate and accelerate the promiscuous yet selective binding of p53-TAD via conformational selection[37,42,43], yet much of its kinetics and the underlying structural dynamics are elusive.

## Results

### NMR reveals rich multi-timescale dynamics of p53-TAD

The timescales and dynamics probed by our combined approach are schematically summarised in Fig. 1B. To access so-called supra-$\tau_c$[44] dynamics of p53-TAD (residues 1–73) in the low µs range, we have utilised recent advances in high-power RD NMR measurements, in particular, amide proton ($^1H_N$) E-CPMG (CPMG: Carr-Purcell-Meiboom-Gill)[45] performed at 1.2 GHz (the part of the red curve indicated by the

purple bar). To detect even faster protein motions, we also measured under supercooled conditions at 263 K[46] (dashed purple bar) and found dynamics around 4 µs, e.g., for residues 19, 24, and 54 (green curves in Fig. 1C, D; all measured residues are shown in Supplementary Information Fig. 1 and the extracted timescales in Supplementary Information Table 1). Assuming Arrhenius behaviour and using Bayesian inference to rigorously estimate uncertainties, these supra-$\tau_c$ dynamics are estimated to take place between 66 ns and 100 ns at room temperature (298 K) (Fig. 1B, middle red arrow, Fig. 1F). Here, the rather large uncertainty (±1 standard deviation around the mean) reflects the narrow temperature window between 262 K and 265 K that can be accessed by these measurements. As an additional independent check, we estimated the room temperature timescale using Eyring's formula[47,48], yielding similar timescales (Arrhenius: 83.6 ± 18.4 ns; Eyring: 63.2 ± 13.8 ns).

Notably, these supra-$\tau_c$ dynamics are much slower than the nanoseconds reconfiguration dynamics previously observed for IDPs, e.g., by room temperature single-molecule FRET measurements[16,20]. These slower dynamics are therefore governed by markedly higher energy barriers, which we will collectively refer to as 'tier 1'. Incidentally, loop-closure dynamics of p53-TAD measured by photoinduced electron transfer fluorescence correlation spectroscopy (PET-FCS)[49] occur at similar timescales, and 50 ns to 100 ns reconfiguration dynamics were also observed for a compact cold shock protein by single molecule FRET/nanosecond FCS[33]. However, because NMR observables are very different from non-local distance fluctuations probed by fluorescence spectroscopy, also the underlying structural dynamics will likely be different. We also observed fast reorientation dynamics (Fig. 1B, right red arrow), governed by lower energy barriers, which we will refer to as 'tier 2'. Figure 1G shows that these two timescales are measured for many residues (green crosses and circles), suggesting collective structural dynamics involving larger protein segments beyond local fluctuations.

### Tier 1: sub-microsecond supra-$\tau_c$ dynamics of p53-TAD

Which structural motions cause the unexpectedly slow tier 1 dynamics of this IDP? As sketched in Fig. 2A, these dynamics might arise either from the folding and unfolding of transient secondary structure elements (e.g. the helices involving residues 18–26 or 40–55)[37] or, alternatively, from the collapse and subsequent dissolving of a hydrophobic patch[50]—or due to other unknown structural dynamics.

Evidence for helix-formation dynamics comes from partial helicity seen in our NMR chemical shift measurements, which agree with previously observed[12] helical populations between 15% and 30% for the residues within helix 1 (Fig. 2C). To test which of the above two processes causes the observed tier 1 dynamics, we also performed additional NMR measurements on the previously described Pro27→Ala mutant (P27A)[43]. Pro27 is a highly conserved residue and its alanine substitution has not been observed in naturally occurring variants. Rather, this mutant was originally designed to remove a "helix breaker" proline residue at the C-terminal end of helix 1, thereby changing its helical propensity[43]. If the tier 1 dynamics were dominated by helix folding and unfolding, one would therefore expect these dynamics to be altered in the P27A mutant. Indeed, the helical population of P27A is increased from 30% to 60% and, probably as a consequence[40], this mutant also shows a markedly increased affinity to MDM2[51].

However, our $^1H$ E-CPMG RD profiles (Fig. 1C–E) measured at 263 K and at 1.2 GHz reveal that the timescale of the tier 1 supra-$\tau_c$ dynamics is the same for WT and P27A mutant. Unexpectedly, additional much slower 215 µs kinetics is detected for the P27A mutant, which is absent in WT p53-TAD (orange curves and symbols in Fig. 1C–E, G; see also Supplementary Information Fig. 3), pointing to distinct additional structural dynamics, different from—and much

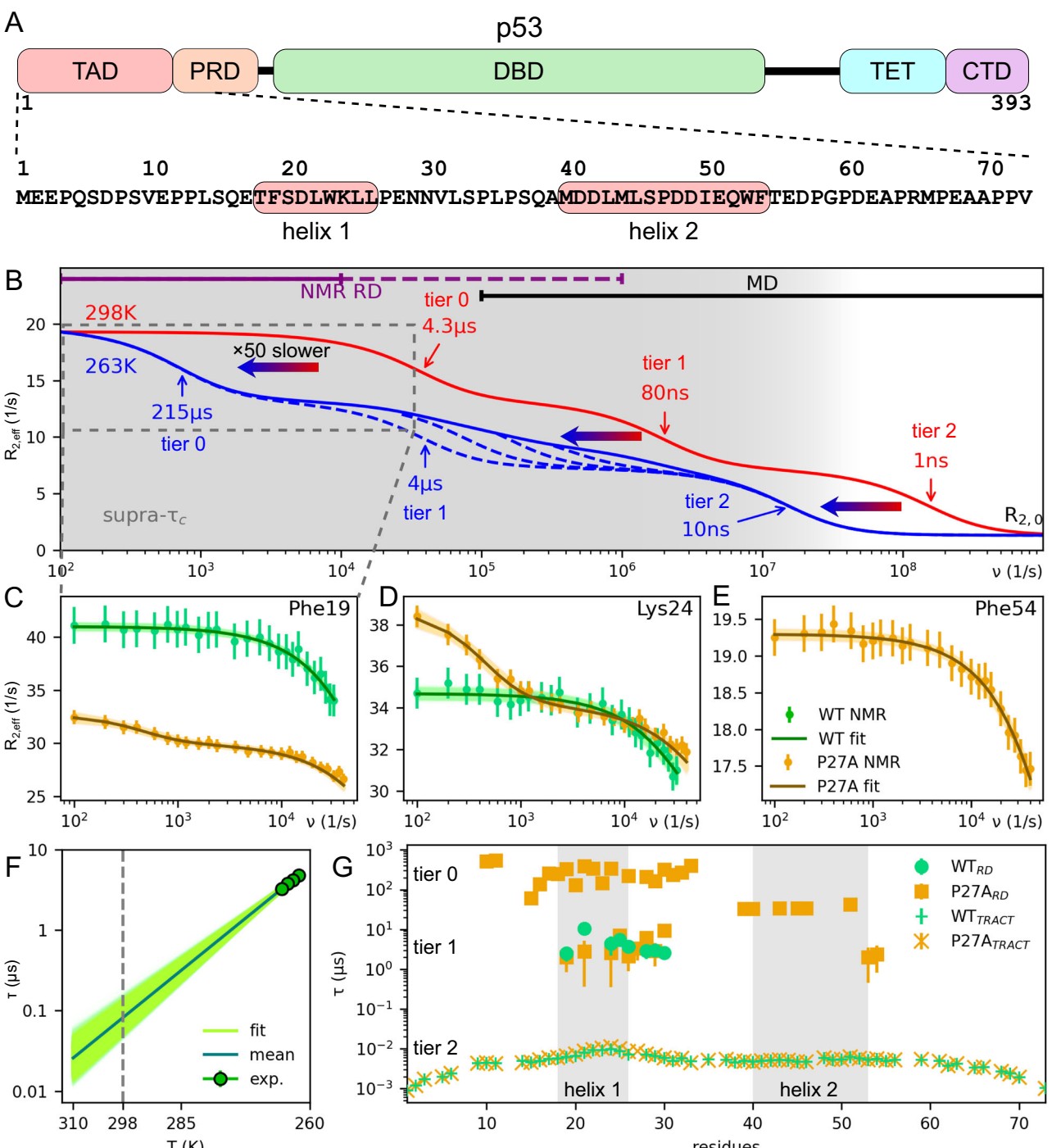

**Fig. 1 | Hierarchical dynamics of p53-TAD across multiple timescales revealed by NMR relaxation dispersion and MD simulations. A** Domain organisation of full-length p53 protein and sequence of the p53-TAD with highlighted helical segments. **B** Schematic representation of RD profiles revealing three distinct tiers of structural dynamics: slow dynamics (215 μs and 4.3 μs, tier 0), fast dynamics (4 μs and 80 ns, extrapolated using Arrhenius' equation, tier 1) and molecular tumbling (10 ns and 1 ns, tier 2) at 263 K (blue) and 298 K (red), respectively. Thin arrows indicate timescales from NMR relaxation experiments scaled by $(2\pi)^{-1}$ for easier interpretation in terms of the well-known Lorentzian function; at the two different temperatures (blue: 263 K, red: 298 K), similar dynamical processes occur at timescales differing by an Arrhenius factor of ca. 50 (horizontal thick arrows, see **F**). Timescales accessible to high-power NMR RD measurement and MD simulations are indicated as purple and black bars; supercooling renders faster timescales accessible to NMR RD (dashed purple bar). The timescale window of the supra-$\tau_c$ dynamics is indicated in grey. **C**–**E** Example mean amide proton ($^1H_N$) CPMG RD

profiles measured at 263 K for three different residues (dots) and fitted CPMG curves (posterior distribution shown as lines) including error bars indicating standard deviation of the measurement. Note that no dispersion could be measured for Phe54 of the WT due to spectral overlap. **F** Temperature dependence of relaxation timescales using supercooled NMR $^1H_N$ $R_{1\rho}$ measurements between 262 K and 265 K (mean as green dots, errors are ($\pm$ SD) within symbol size) and Arrhenius extrapolation to 298 K (mean as green line, uncertainty shown by the posterior distribution in light green). **G** Timescales obtained at 263 K for all measured residues by fitting RD data as shown in (**C**–**E**). WT RD profiles (green circles) show a single timescale; several P27A mutant RD profiles show two separate timescales (orange squares), one of which (4 μs) is similar to WT; shown are only those residues for which relaxation dispersion was detected. All residue level tumbling timescales (crosses) are similar for WT and P27A. Error bars are within symbol size; grey shaded areas indicate helix 1 and helix 2.

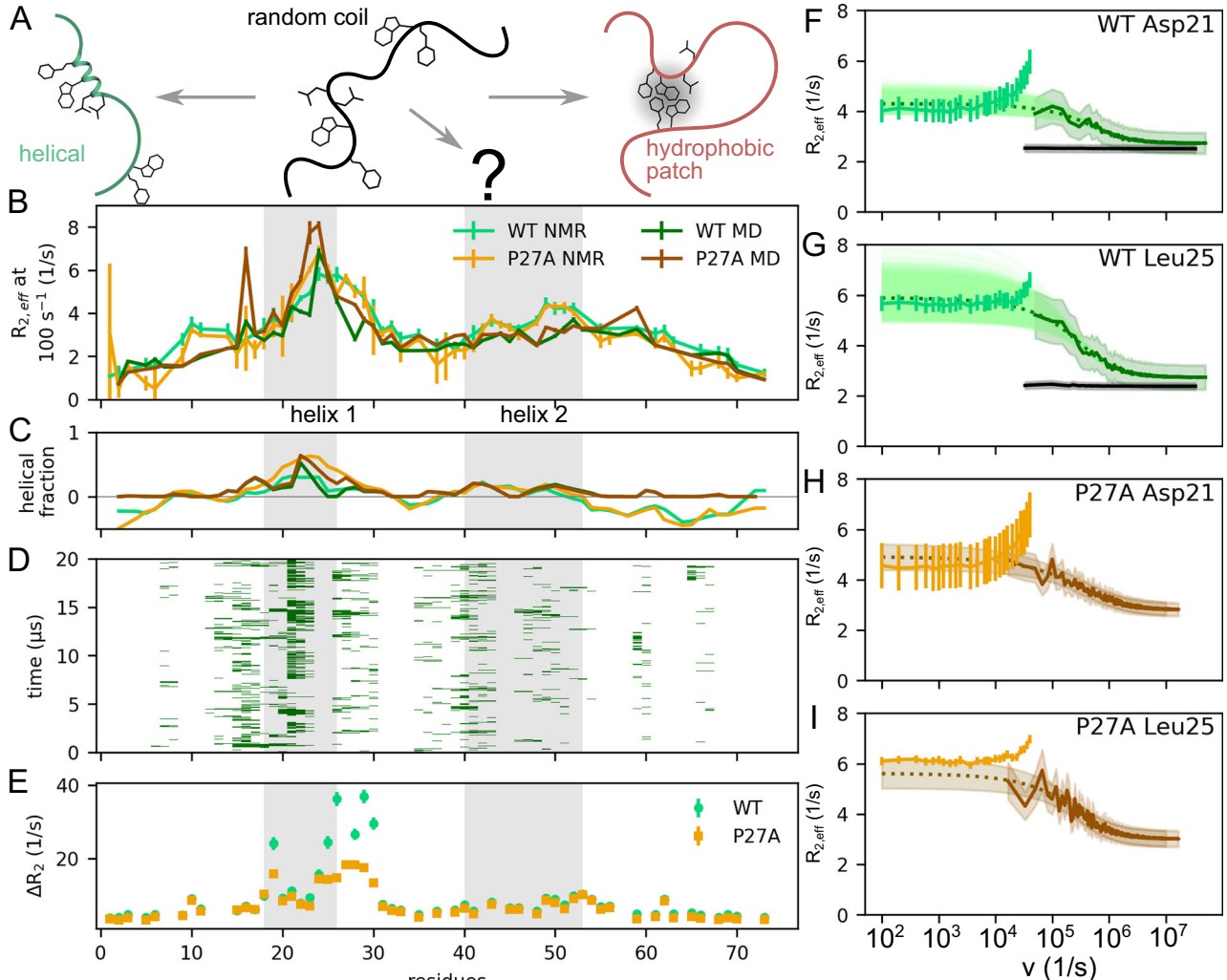

**Fig. 2 | Tier 1 supra-$\tau_c$ dynamics of p53-TAD as observed by MD simulations at 298 K and compared to NMR RD measurements. A** Schematic representation of two plausible structural origins of the observed supra-$\tau_c$ (tier 1) dynamics: folding/unfolding of transient helices and collapse/dissolution of hydrophobic clusters. **B** Comparison of amplitudes of 298 K NMR RD profiles at $100\,\text{s}^{-1}$ CPMG frequency for each measured residue with those amplitudes extrapolated from MD simulations (darker colours) for WT (green) and P27A mutant (orange). Experimental and statistical uncertainties are shown as error bars ($n_{\text{experiment}} = 2$; $n_{\text{MD}} = 30$); grey shaded areas indicate helix 1 (18−26) and helix 2 (40−54). **C** Comparison of the helix fraction of each residue estimated from measured NMR chemical shifts and SSP with those derived from the MD simulations using dihedral angles (colours as in **B**). Negative values indicate a preference for extended conformations. **D** Dynamics of helix formation; green bars indicate instances during a representative 20 µs MD simulation at which each residue and its neighbours form a short helical segment. **E** Relaxation amplitude difference $\Delta R_2$ between the highest available CPMG frequency and that of relaxation due to overall molecular tumbling, as measured via TRACT experiments. Differences $\Delta R_2$ are shown for the WT (green dots) and the P27A mutant (orange squares). All measurements were performed at 263 K at a magnetic field strength of 1.2 GHz. Both NMR experiments used 2 repeated measurements. **F−I** Four examples of RD profiles (lighter solid lines) measured at 298 K and those calculated from the MD simulations (darker solid) and extrapolated to the NMR frequency range (dotted). As a control, black lines show RD profiles calculated from WT simulations for which unfolding of the α-helix was suppressed. Uncertainties of the measured profiles (vertical bars) were estimated from repeated measurements; for the calculated RD profiles, standard deviations around the mean are indicated as shaded areas.

slower than−tier 1. We will refer to these new dynamics as "tier 0", the structural origin of which is elusive at this stage.

Notably, because at each particular CPMG frequency the measured $R_{2,\text{eff}}$ value contains relaxation contributions from all faster relaxation processes, the RD profiles shown in Fig. 1C−E (and in Supplementary Information Figs. 1, 4 and 5) also contain information about these faster conformational dynamics, even though these occur far outside the RD detectable timescale window between the rotational correlation time and ~4 µs. To disentangle the contributions from these faster kinetics from the components of the RD profiles that arise from exchange processes at the slower tier 0 relaxation frequencies, we independently determined residue-specific rotational correlation times $\tau_c$ via TRACT[52] (TROSY for rotational correlation times)

measurements (crosses in Fig. 1G). Accordingly, the difference $\Delta R_2$ between the $R_{2,\text{eff}}$ value measured at the highest CPMG frequency and the TRACT-derived intrinsic transverse relaxation rate $R_{2,0}$ quantifies the contribution from additional supra-$\tau_c$ dynamics that are faster than what could be detected at the maximum E-CPMG frequency (Fig. 2E). This finding provides an independent approach to detect dynamics faster than the timescale accessible to the E-CPMG RD measurements.

To isolate these fast contributions quantitatively, we subtracted the TRACT-derived relaxation rates from the RD measurements acquired at 1.2 GHz (Fig. 2E). Indeed, residues within and adjacent to helix 1 (residues 18–30) show markedly enhanced $\Delta R_2$, indicating the presence of dynamics that are slower than overall tumbling ($\tau_c$) yet faster than those directly captured by the E-CPMG RD experiments.

Although the precise timescale of these structural dynamics cannot be determined from these NMR experiments alone, they clearly establish the existence of supra-$\tau_c$ tier 1 dynamics within this otherwise "blind spot" of NMR RD measurements.

## Unbiased atomistic simulations validated against NMR

To reveal the structural motions that cause the observed dynamics on all three tiers identified by NMR, we have carried out extensive MD simulations of both p53-TAD WT and P27A mutant in explicit solvent at 298 K. Contrary to previous studies[53,54], these simulations are unbiased and have not been fitted to any p53 measurement, which allowed assessing their accuracy by calculating RD profiles, i.e., effective relaxation rates $R_{2,\text{eff}}$, from the MD simulations. For each residue, two main components contribute to $R_{2,\text{eff}}$, (1) relaxation arising from exchange between states (e.g., folding and unfolding) with different chemical shifts $R_{2,\text{ex}}$; (2) relaxation caused by molecular tumbling and sub-$\tau_c$ internal motion summarised in $R_{2,0}$. The effective relaxation $R_{2,\text{eff}}$ contains both components $R_{2,\text{ex}}$ and $R_{2,0}$, which therefore were calculated separately from the trajectories. We note that due to the rich internal dynamics of this IDP, and in contrast to folded proteins, no well-defined overall tumbling timescale exists, which is why we resorted to a residue-wise analysis. Owing to the total MD trajectory length of 2.4 ms, the calculated RD profiles cover slow timescales up to 10 μs (sketched in Fig. 1B, black bar), thus enabling direct comparison with the experimentally determined ones.

To enable a direct comparison between RD profiles computed from our MD simulations and those measured by our NMR experiments, we measured full RD profiles also at 298 K. Figure 2F–I shows representative comparisons between the measured RD profiles (solid, lighter colours) and those calculated from the MD simulations (solid, darker colours) for two selected residues of both the WT (Fig. 2F, G) and the P27A mutant (Fig. 2H, I). Complete comparisons for all analysed residues are provided in Supplementary Information Figs. 4 and 5. Note that at higher CPMG frequencies, the measured RD profiles exhibit an unphysical increase, attributable to high-power artifacts likely arising from hardware limitations rather than sample heating. Therefore, the comparison between experiment and simulation was restricted to the low CPMG-frequency region of the RD profiles, which are free of such artifacts and reliably report on exchange dynamics.

To enable the most direct comparison, we fitted the RD profiles calculated from the MD simulations using Bayesian inference and a stretched CPMG model[55] (dotted lines), without using the measured RD profiles. As described in more detail in Supplementary Information Fig. 6, this stretched CPMG model generalises the established two-state model[56] to many states and, hence, represents a superposition of many relaxation rates. Very good agreement is seen, e.g. for residue Asp21 (Fig. 2F, H), whereas Leu25 (Fig. 2I) illustrates a typical deviation. Figure 2B compares the low frequency $R_{2,\text{eff}}$ amplitude measured for all residues by NMR with those obtained from the MD simulations via the stretched CPMG fits.

Overall, good agreement is achieved for most residues, with an average deviation of ~15%. In particular, the dynamic profiles in Fig. 2B, showing higher amplitudes for the helical regions, are very similar in shape for both measured and calculated amplitudes. Deviations are seen mainly for the N-terminal region up to residue Val10, likely caused by the presence of 2–4 additional residues in the experiments, which were required for cloning. These were omitted in the MD simulations, which aimed at an accurate modelling of the WT p53-TAD. Indeed, after inclusion of these additional residues within the MD simulations, much better agreement is seen also for the N-terminal region, as shown in Supplementary Information Fig. 7. The deviation seen for Asn28 is presumably due to the absence of proline isomers in the MD simulations, supported by the fact that much better agreement is seen for the P27A mutant (Fig. 2B, orange). Further, the helical fractions for all

residues calculated from the MD simulations (Fig. 2C, darker colours) also agree well with those derived from the NMR chemical shifts using SSP[57] (lighter colours), see also Supplementary Information Fig. 8.

We note that the comparison (Fig. 2B, F–I) between the measured RD profiles and those calculated from MD simulations involves several sources of uncertainty. First, IDPs are known to be particularly sensitive to force field inaccuracies[58–60]; we have therefore carried out test simulations using different force fields and obtained similar profiles for the NMR observables (Supplementary Information Fig. 9). The best agreement was seen for the Amber99sbws force field, which we therefore chose for further analysis.

Second, despite milliseconds of sampling, the MD ensemble may not fully cover the full structural ensemble probed in the experiments. To assess convergence of our MD simulations, we compared observables calculated separately from three consecutive 20 μs blocks of the 60 μs long P27A trajectories. Chemical shift-based RD profiles (without the $R_{2,0}$ tumbling related offset) were calculated for the full trajectories as described above and were averaged over 30 independent trajectories using the first, second and third 20 μs block of each trajectory (Supplementary Information Fig. 10). As can be seen, most averaged RD profiles are within their respective error ranges except those of residues Asp49, Phe54, Glu56, Glu62, Ala63, and Met66. The RD profiles of these residues, therefore, might not be sufficiently converged, and therefore their relaxation times were omitted from Fig. 2B.

Lastly, the accuracy of calculated RD profiles relies on that of the chemical shift calculations using SPARTA+[61]. To assess the accuracy of these empirical predictions, we have also calculated chemical shifts using SHIFTX2[62] and obtained almost identical values, deviating by ca. 2.5% on average. Next, we compared calculated chemical shifts, averaged over our MD trajectories, to our own NMR measurements as well as to chemical shifts by Wong et al.[63] (Supplementary Information Fig. 8). For the $C_\alpha$ chemical shifts, very good agreement is seen, with Pearson correlations of 0.996 and a mean absolute error of 0.26 ppm and 0.24 ppm (Supplementary Information Fig. 8A, C), respectively. Proton chemical shifts, from which RD profiles were calculated, show a Pearson correlation of over 0.76 and a mean absolute error of 0.13 ppm (Supplementary Information Fig. 8B, D). We note that the amplitudes of the RD spectra are sensitive to the somewhat larger uncertainty of the predicted proton chemical shifts and, therefore, the comparison of calculated to measured RD profiles provides an independent accuracy assessment. In contrast, relaxation times derived from calculated RD spectra are insensitive to such inaccuracies.

Notably, because the orientational correlation times were directly calculated from the orientational autocorrelation of the backbone NH bond vector without recourse to chemical shifts, their comparison with the measured TRACT relaxation times also provides an independent test of the accuracy of the simulations. Indeed, for all residues except the N-terminal ones, very good agreement is seen (Supplementary Information Fig. 11, $R_2$ panel). Here, too, this deviation is mainly due to the additional residues required for sample preparation, as evidenced by the markedly improved agreement obtained for additional control simulations which included these residues (Supplementary Information Fig. 7). Finally, deviations between experiment and MD simulations might arise from the very slow isomerisation dynamics of prolines[64], which are not described by our simulations. However, additional control simulations comparing, e.g., WT cis-Pro8 with trans-Pro8 showed no significant differences (Supplementary Information Fig. 11).

To further assess the accuracy of our unbiased MD simulations, we have calculated from our trajectories all observables derived from further independent calculations and experiments collected from the literature. These were ensemble averaged size determined by size-exclusion chromatography, small-angle X-ray scattering, and dynamic light scattering (Supplementary Information Fig. 12), as well

as distances determined by fluorescence resonance energy transfer (Supplementary Information Fig. 13), photoinduced electron transfer fluorescence correlation spectroscopy (Supplementary Information Fig. 14), and paramagnetic relaxation enhancement (Supplementary Information Fig. 15). As discussed in more detail in these Supplementary Information Sections, good agreement is seen for all these observables.

## MD simulations reveal tier 1 helix-formation dynamics with multiple relaxation rates

Taken together, these quantitative comparisons of our atomistic simulations to a broad range of measurements suggest that the simulations describe the structural dynamics of p53-TAD sufficiently accurately to allow identification of the structural motions that give rise to the observed tier 1 dynamics. To this end, Fig. 2D shows the secondary structure dynamics particularly within the helical regions (grey), revealing sub-μs folding and unfolding of short helical structure elements (green bars). These dynamics are indeed the dominant cause of the observed NMR RD dynamics, as evidenced by control simulations, for which the α-helical regions have been forced to remain in helical geometry. Indeed, the RD profiles calculated from these control simulations (black curves in Fig. 2F, G) do not show any tier 1 dynamics. These results establish sub-μs helix folding dynamics as the structural determinant of tier 1, on top of the known and much faster tier 2 structural reorganisation dynamics[12,20] (cf. Fig. 1B).

As can also be seen in Fig. 2D, these secondary structure folding dynamics are quite complex and are neither described by a fully cooperative two-state model, nor are the partial folding steps fully uncorrelated. Rather, these tier 1 dynamics seem to involve transitions between many different partially helical conformations and, accordingly, also involve many different timescales.

To characterise these partially cooperative tier 1 dynamics in more detail, we have used the WT MD trajectories to derive a Markov state model of the helix 1 conformational dynamics (Fig. 3). Figure 3A, B shows the MD ensemble after dimension reduction via time-lagged independent component analysis (TICA)[65]; each dot represents a structure snapshot, characterised by the most relevant independent components IC1 and IC2, which describe collective motions. In Fig. 3A, each of the obtained seven Markov states (colours and black numbers) represents a particular conformation of the helix, characterised in Fig. 3B by the number of intra-helical hydrogen-bonds (colours) ranging from ten (fully folded) to zero (unfolded). The conformations corresponding to these seven Markov states (top of Fig. 3C, hydrogen-bonds are shown as red dots) also display this sequence of decreasing helical structure. Our Markov state analysis yields free energy estimates for each state (bottom of Fig. 3C, thick bars) and transition rates between these states, ranging from 0.2 to 24.1 μs (arrows). These form a 'folding funnel' very similar to the one characteristic of folded proteins[7]. Resolved by residue, the whole spectrum of timescales of the p53 dynamics (Fig. 3D, blue bars) densely covers a range between a few and several hundred ns. Notably, here the Markov state model is not used to extrapolate to larger implied timescale otherwise inaccessible to the simulations[30]; rather, these dynamics are fully sampled by our simulations, and the Markov state model only serves to extract their rates and structural intermediates from the simulations.

Taken together, the IDP undergoes structural dynamics with transient and stepwise secondary structure formation that are far more complex than a simple two-state model. This finding also explains why a single two-state RD profile[56] is unable to properly describe the E-CPMG RD profiles calculated from our MD simulations; instead, a stretched CPMG model that represents a superposition of many CPMG curves at different frequencies[55] (indicated by the blue dashed lines in Fig. 1B), fits the calculated profiles very well (Fig. 2 and Supplementary Information Figs. 4 and 5). As illustrated in

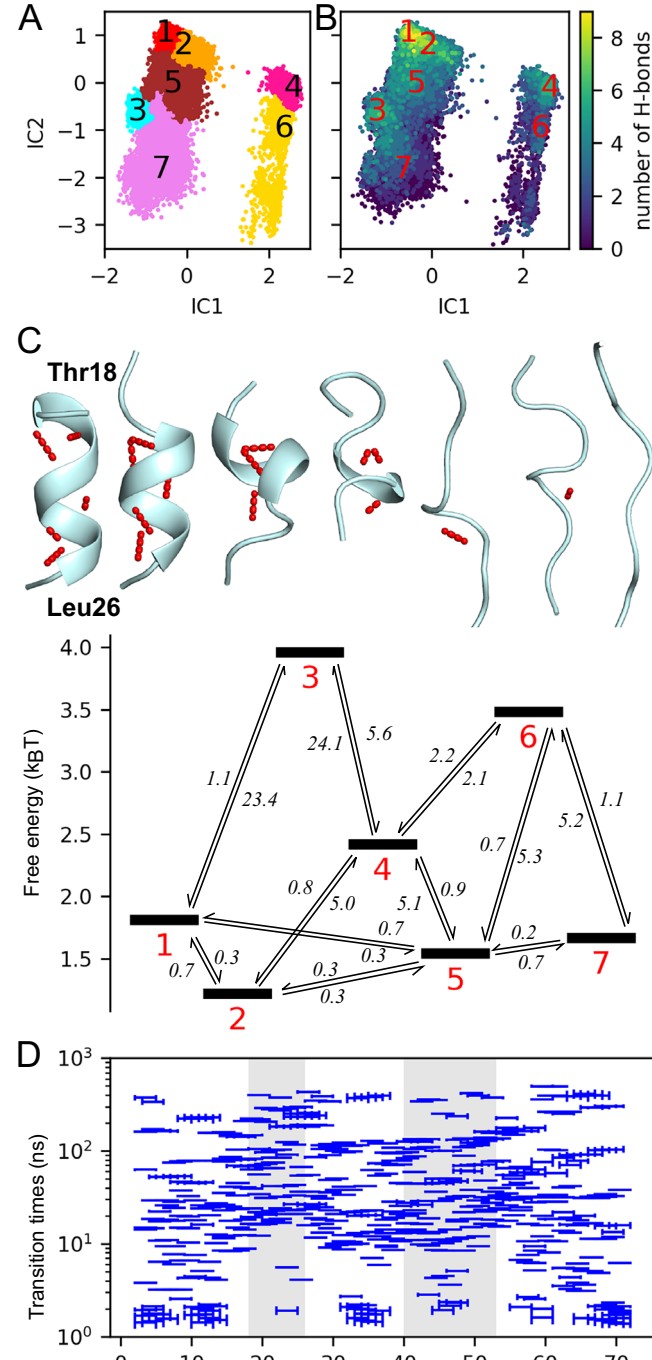

**Fig. 3 | Markov state model reveals multi-timescale folding/unfolding dynamics of helix 1 underlying tier 1 motions of p53-TAD. A** Projection of the conformational ensemble of helix 1 (residues 18–26) onto the TICA space defined by the two collective coordinates IC 1 and IC 2 that contribute most to the intra-helical dynamics using WT trajectories. Colours and numbers indicate seven Markov states. **B** The same projection as in (**A**), but coloured by the number of intra-helical hydrogen bonds. **C** Representative structures of the seven Markov states (intra-helical hydrogen bonds shown as red dots, top) and their free energies (black bars, below); mean first passage times of transitions between the states (arrows) are shown in μs. For a similar analysis (**A**–**C**) of helix 2, see Supplementary Information Fig. 16. **D** Per-residue timescale spectra (blue bars) calculated from MD trajectories, obtained using four residue (one helix turn) segments across the p53-TAD sequence (statistical uncertainties (±SD) shown as error bars). Grey shaded areas indicate helix 1 and helix 2 residues.

Supplementary Information Fig. 6 and described in more detail in the respective Supplementary Information Section, the stretching factor of approximately $\gamma = 0.6$ obtained from these fits is consistent with a timescale spectrum covering a similar range as the transition times shown in Fig. 3C. In a similar spirit as the stretched exponential rebinding kinetics originally proposed by Frauenfelder[5] and more recently adapted by Palmer[24], Edholm and Blomberg[66], and Blackledge[67], the broader spectrum of many relaxation rates described by the stretched CPMG model, can be understood as arising from a multi-state Markov model of interconverting conformational states which are separated by energy barriers of different heights. Specifically, the relaxation rates are the eigenvalues of the master equation that defines the Markov model[68].

**Tier 0: multi-μs dynamics due to transient formation of tertiary structure elements resembling folding intermediates of natively folded proteins**

Which structural motions give rise to the slow 215 μs (tier 0) kinetics revealed by the NMR RD profiles of the P27A mutant (indicated by the left side of the blue curve in Fig. 1B)? To answer this question, we systematically searched for long-lived stable structures in our MD trajectories. Indeed, a residue-resolved analysis of $C_\alpha$-$C_\alpha$ distance fluctuations (Fig. 4A) revealed sporadic multi-μs collective excursions of pronounced stability, which are both larger and longer lived than the transient helices (Fig. 2D) of tier 1. We identified a total of 18 such long-lived events in the 0.6 ms WT simulations and 75 for P27A, three of which are shown in Fig. 4A. They represent metastable tertiary structure elements larger than simple α-helices, which persist up to 5 μs and are mostly stabilised by hydrogen bonds. We note that the observed μs persistence times cannot be quantitatively translated into kinetic rates, because rare and longer-lived tertiary structures may have been missed by our simulations. Therefore, the actual formation rates may be slower.

For the P27A mutant, the simulations suggest that the removal of the sterically restrictive proline increases the conformational flexibility of p53-TAD, which enables the formation of larger and longer-lived tertiary structures. The larger number of long-lived structure formation events, along with their increased total lifetime seen in the P27A simulations, explains the much more pronounced low-frequency signal seen in the P27A CPMG RD profiles.

Some of these structures are also stabilised through hydrophobic contacts (e.g. the rightmost example in Fig. 4A); hence, and in contrast to the faster tier 1 dynamics, hydrophobic collapses seem to contribute to the tier 0 dynamics. Notably, these long-lived tertiary structures frequently form in the presence of (and involve) one of the two otherwise much shorter-lived α-helices (Fig. 4B and, e.g. the largest structure shown in Fig. 4A), thereby stabilising these helices over much longer timescales. This observation suggests that, at least for this particular IDP, the α-helices form "nucleation seeds", which promote subsequent tertiary structure formation attempts, similar to primary folding intermediates seen for natively folded proteins[69,70].

This effect is particularly notable for tertiary structures involving residues 45–65, where only the first few residues located in the N-terminal segment are involved in helix formation (Fig. 4B). In contrast, the terminal regions of the protein exhibit only fast reorganisation dynamics, whereas there are also segments outside the helix 1 and helix 2 regions (e.g. residues 10–35 particularly for P27A) for which formation of transient tertiary structures associated with slow dynamics are seen. This finding also indicates that secondary structure formation is not a strict prerequisite for transient tertiary formation and the associated slow tier-0 dynamics. This pronounced sequence and structure dependent heterogeneity adds to the complexity of the underlying hierarchical free energy landscape of this IDP.

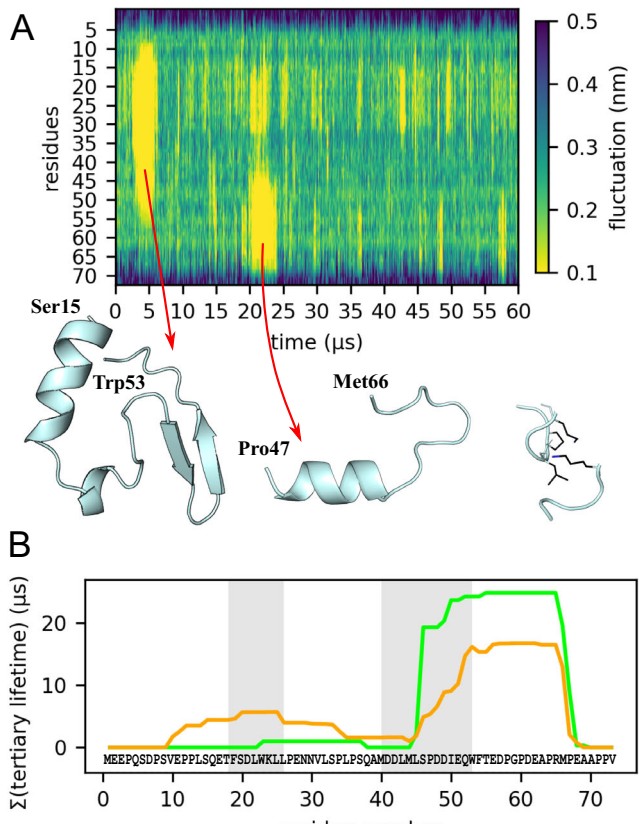

**Fig. 4 | Residue distance fluctuation analysis reveals stable tertiary structures mainly around transient helices. A** Example of $C_\alpha$-$C_\alpha$ distance fluctuations analysis. The trajectory visited multiple transient tertiary structures from which two are shown. For every residue, distance fluctuations, ranging from 0.1 (yellow, rigid) to 0.5 nm (dark blue, flexible), were determined and averaged over a 0.1 μs sliding time window. **B** Accumulated tertiary structural element lifetimes of all WT (green) and mutant (orange) MD trajectories, normalised by trajectory length to facilitate comparison. For the WT, particularly for the helix 2 region, these lifetimes are dominated by very few long-lived tertiary structures and, therefore, are subject to large statistical uncertainty. Grey shaded areas indicate helix 1 and helix 2 residues.

**Tier 2: a stretched spectral density function reflects complex multistate dynamics also at fast nanosecond timescales**

Contributing to the NMR RD profiles are also faster, ns and ps motions, which have been probed by regular relaxation experiments[24] and here add an offset to $R_{2,\text{eff}}$. These motions are characterised by the spectral density function (SDF), which is the Fourier transform of the correlation function of the dipolar couplings and the chemical shift anisotropies[71]. Despite its central role, the full SDF is inaccessible to experiments; rather, only isolated points of the SDF at specific frequencies can be determined, using combinations of multiple NMR measurements such as $\eta_{xy}$, $\tau_c$, $R_1$, $R_2$, and NOE at different conditions[10].

The considerable length of our MD trajectories enabled us to calculate full SDFs of p53-TAD. Figure 5A shows examples for four residues, two of which are also shown in Fig. 2; SDFs for all residues are shown in Supplementary Information Fig. 17. For each residue, the SDF was calculated over a broad frequency range covering more than four orders of magnitude (green lines). Notably, the SDF calculated from our MD simulations are logarithmically stretched relative to the Lorentzian functions that would result from a single $\tau_c$[24], which we refer to as "simple model" (blue lines), as evident from the shallower slope at higher frequencies. In this simple model (Eq. (7)), the underlying dynamics are described by a single tumbling timescale $\tau_c$, which we measured by the TRACT experiment described above. This marked logarithmic stretching was observed for all residues and is in line with a

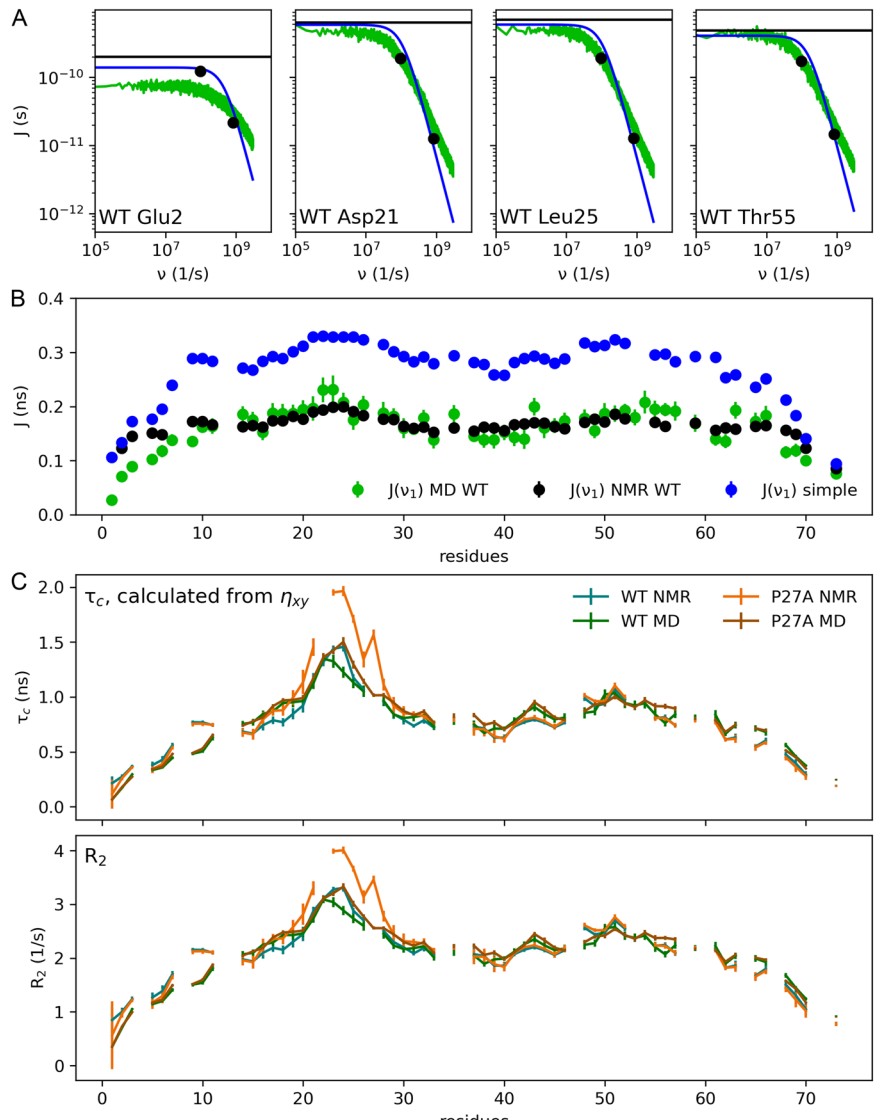

**Fig. 5 | Comparison between calculated spectral density functions (SDFs) $J(v)$ with NMR measurements at 298 K. A** Examples of SDFs calculated from our MD simulations (green lines) and measured SDF values at frequency $v = 0$ (horizontal black lines) as well as at frequencies $v_1 = \omega_N/2\pi = 96.3$ MHz and $v_2 = \varepsilon\omega_H/2\pi = 826.5$ MHz, respectively (black dots) for the residues of the p53-TAD shown in Fig. 2F–I and for two additional residues of the N-terminus and of helix 2. Blue lines show SDFs calculated from a simple two-state model. Per literature convention, frequencies $v$ are given in units 1/s, whereas frequencies $\omega$ are in units rad/s. To facilitate easier comparison with low-frequency spectra from CPMG experiments, we here show high frequency spectra such as the SDFs also as a function of $v$ and in

units of 1/s. **B** Measured SDFs at $v_1 = \omega_N/2\pi = 96.3$ MHz ($J(v_1)$, black dots, $n = 2$) compared to values calculated from our MD simulations (green, $n = 30$) and predicted from a simple two-state model (blue); Data are presented as mean values ± SEM and are smaller than the symbols. **C** Comparison of $\tau_c$ and the rate constant $R_2$ derived from NMR measurements at 1.2 GHz ($n = 2$) with values calculated from MD simulations ($n = 30$) for the p53-TAD WT (turquoise and green lines) and the P27A mutant (orange and brown lines). For proline residues without backbone amide protons, no spectra were obtained. Complete results are provided in the Supplementary Information Figs. 11, 17 and 18. Data are presented as mean values ± SEM.

similar observation for RD profiles (Supplementary Information Fig. 6), which strongly suggests that also the tier 2 reorientation dynamics (which determine the SDF) comprise not one but many timescales. A plausible explanation could be that tumbling motions of certain residues depend on their immediate environment and are, e.g., slower when in a transient helical conformation than in a disordered state.

For comparison to the experiment, we determined SDFs $J(0)$, $J(\omega_N)$, and $J(\varepsilon\omega_H)$ at three distinct frequencies (black line and dots in Fig. 5A and Supplementary Information Fig. 17) using multiple NMR measurements at various conditions and using $\varepsilon = 0.87$ as described earlier by Farrow et al.[72] As can be seen, these agree much better with the SDF values calculated from our MD simulations than with the simple model, as is also evident for all other residues (Supplementary

Information Fig. 18). Notably, at $v_1 = 96.3$ MHz (Fig. 5B), calculated and measured values agree very well, whereas the simple model (blue) consistently predicts larger values, due to the different slopes of the high-frequency parts, underscoring the multi-timescale reorientation dynamics of p53-TAD within tier 2. At the highest frequency ($v = 826.5$ MHz, Supplementary Information Fig. 18C), the simple model predictions agree better with the measured values, which, however, we consider largely coincidental, due to the fact that this frequency is often close to the intersection point of the two spectra (cf. Supplementary Information Fig. 17). While the SDF profile $J(v_2)$ calculated from our MD simulations has a quite similar shape as the measured one, the values are consistently too large. The reason for this discrepancy remains unclear; because the nanosecond dynamics

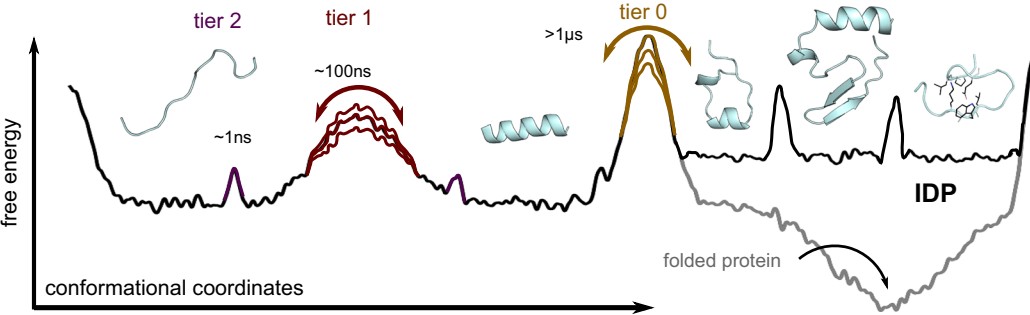

**Fig. 6 | Sketch of the p53-TAD IDP hierarchical free energy landscape emerging from our combined NMR/MD study.** Schematic free energy landscape (black curve) summarising the three distinct tiers of p53-TAD dynamics emerging from our combined NMR/MD study. A hierarchy of free energy barriers (coloured) similar to that known for folded proteins[5] governs fast (ns) reorientation and reconfiguration dynamics (tier 2 in purple), slower (sub-μs) supra-$\tau_c$ dynamics arising from transient helix formation (tier 1 in red), and very slow (multi-μs to ms) formation of transient tertiary structure elements (tier 0 in beige). The indicated variation of barrier heights is also similar to that observed for folded proteins and gives rise to a correspondingly broadened spectrum of relaxation times, which explains the observed stretched RD profiles. Contrary to folded proteins (grey curve), the transient tertiary structure elements of IDPs (tier 0) are only metastable and difficult to access experimentally. Structures shown as ribbon plots within the respective energy wells indicate typical intermediates seen in the MD ensemble.

produced by the Amber99sbws force field is known to be slightly too slow (cf. Table 2 in ref. 73), we speculate that equipartition might imply correspondingly larger amplitudes at these fast frequencies. In addition to this systematic deviation, larger differences are seen for the N-terminal residues up to Val10, again likely due to the few residues added during cloning, which were absent in the simulations.

A more direct comparison to NMR is achieved by calculating the above five NMR observables $\eta_{xy}$, $\tau_c$, $R_1$, $R_2$, and NOE, from the MD simulations (Fig. 5C shows $\tau_c$ and $R_2$; all five observables are shown in Supplementary Information Fig. 11). Overall, good agreement is seen, facilitating structural interpretation of the parts of the SDFs that are inaccessible to experiment. Deviations are seen for mutant residues particularly in the region 23–27, which is most likely due to the pronounced tier 0 dynamics of P27A. Whereas we used the longest possible autocorrelation window size (10 μs) for the calculation of the SDFs, the experimental timeframe is much wider and thus can detect (and be affected by) slower events. Notably, the SDF calculated from the simple model generally deviates more from the experiment at $\nu = 96.3$ MHz than the SDF calculated from our simulations, providing further and independent evidence for the presence of multi-state dynamics covering a wide spectrum of timescales also within this fast tier 2 regime. Indeed, such fast motions have been observed for different IDPs previously, such as librational motions of amide groups, fast backbone conformational fluctuations, and slow chain segmental motions[67,74], which all can contribute to the measured relaxations.

Because the orientational dynamics within secondary structure elements are expected to be slower than those of disordered regions, residues that are involved in the formation of transient helices should show correspondingly slower correlation times. Indeed, whereas fast sub-ns dynamics are seen for the terminal regions, the local reorientation dynamics of residues with substantial helical population (grey shaded areas Fig. 2) slow down to 1 to 2 ns (Fig. 5C, panel $\tau_c$). These timescales agree well with previous NMR measurements[12] as well as with single-molecule FRET experiments on the Csp*Tm* cold-shock protein[16], underscoring the slower rigid-body motion of these transient helices within the faster rearranging disordered phase.

Our simulations also enable comparison of the fast (tier 2) conformational dynamics of p53 to polymer models[21,54,75]. In particular, p53 is highly and rather homogeneously flexible with a persistence length of 2–3 amino acids (Supplementary Information Fig. 19A, B). Its scaling exponent of $\nu = 0.66$ is compatible with a self-avoiding chain model[76], which agrees with previous measurements of polymers in good solvents[77] and is above the critical Θ-point, at which chain-chain and chain-solvent interaction balance at the thermodynamic phase boundary[78].

Exponential fits to the orientation autocorrelation function (Supplementary Information Fig. 19D) show even higher flexibilities at short contour lengths, in line with single-molecule FRET experiments[16]. An unexpected additional component is seen with a considerably longer persistence length of over 5 nm, pointing to so far unresolved partial long-range order, likely due to self-crowding governed by the slower tier 1 and tier 0 structural dynamics. These findings are supported by similar fast components of the time-lagged structure RMSD decay as well as time autocorrelation functions of the radius of gyration and end-to-end distance (10 ns, 22 ns, and 18.5 ns, respectively, Supplementary Information Fig. 19E, F). The latter two also reveal additional, much slower dynamics of 405 ns and 350 ns, respectively, likely arising from tier 1 dynamics. Overall, the structural dynamics of p53 resemble those of a highly flexible heteropolymer with fast reorganisation dynamics[77] at short contour lengths and additional slower dynamics at larger length scales that give rise to pronounced long-range correlations.

## Discussion

Our high-power RD NMR measurements (E-CPMG)[45] performed at 1.2 GHz combined with milliseconds atomistic MD simulations revealed a hierarchy of complex structural dynamics of the intrinsically disordered transactivation domain of the central gene regulation protein p53 (p53-TAD), which is known to form transient α-helices[12] that are relevant for binding and molecular recognition[23,35]. Covering a dynamic range of over seven orders of magnitude, from ps to sub-ms, we found three tiers of quite diverse structural dynamics (Fig. 6). This finding follows independently and consistently from our NMR measurements as well as from our MD simulations. Combined, our results markedly extend the timescale range at which IDP structural dynamics is observed at atomic scales, add structural and dynamic detail to transient structure formation in IDPs[20,22], and thereby further challenge the earlier notion that the lack of stable folded structures of IDPs implies a shallow or weakly funnelled free energy landscape that gives rise only to rather simple and fast tier 2 reorganisation dynamics with occasional secondary structure[17,18,20].

Instead, our findings suggest that the classic Frauenfelder stretched exponential description of folded protein dynamics, which arises from an underlying multi-tier hierarchical free energy landscape[5], also applies to the largely unstructured IDP p53. Notably, we have obtained similar results for Measles Virus N$_{TAIL}$ (Supplementary Information Fig. 20), an unrelated IDP. N$_{TAIL}$ is a pre-molten globule type IDP[79] with little secondary structure except for a short, transiently helical molecular recognition element (MoRE) that forms a stable four-helix bundle when binding the Measles phosphoprotein X-domain[80]. Recent PET

experiments revealed $N_{TAIL}$ dynamics that markedly deviate from simple homopolymer model predictions, which can be explained by the presence of transient tertiary structures that have been observed in recent molecular dynamics simulations[81]. Combined, our findings suggest that these unexpectedly complex hierarchical dynamics may be a feature of other IDPs.

In particular, transient secondary structure formation dynamics are seen on a broad range of supra-$\tau_c$ (10s to 100s of ns) timescales between several ns and sub-$\mu$s, forming a second dynamics level, tier 1. In our simulations, these serve as dynamic precursors for the tier 0 formation of metastable and quite diverse tertiary structures at different positions along the TAD sequence. We speculate that these may be a key factor in the ability of p53 to fold into a broad range of different structures upon binding to a similarly broad range of different partners. Indeed, a structure similarity search in the Protein Data Bank (PDB)[82] revealed that motifs of these tertiary structures seen in our MD simulations also occur in proteins (Supplementary Information Fig. 21). Because such rich conformational selection dynamics has the potential to accelerate binding, this third dynamics tier, emerging from an unexpectedly complex free energy landscape (Fig. 6), may thus be not only a feature of p53-TAD, but also a key to understanding the binding promiscuity of many other IDPs.

Adding further complexity, this timescale hierarchy is expressed at varying degrees along the protein sequence. In particular, whereas the dynamics of the terminal segments are dominated by tier 2 dynamics, tier 1 dynamics are more restricted to the helical regions. Although these tier 1 dynamics serve as precursors for tier 0 dynamics in these regions, transient tertiary structure formation is also observed independent of helical structures, e.g., for residues 56–65 (Fig. 4B).

Our results show that the structural dynamics of IDPs are not limited to fast and essentially random reorganisations, but are organised across a broad hierarchy of timescales, as is characteristic for the conformational dynamics of natively folded proteins. This complexity refines the notion of an inverted free energy landscape[83] and highlights similarities to the (un-)folding dynamics of natively folded proteins, where both MD simulations[84] and Φ-value measurements[85] have revealed transiently stable secondary and tertiary structure elements along unfolding pathways, similar to those seen here for p53-TAD in equilibrium. Knowledge of such transient structures may help shift their conformational ensembles and modulate their activity, such that these structures may become potential drug targets. Unbiased MD ensembles such as the one presented here, which are consistent with NMR and other measurements, may also provide valuable training data sets for deep learning generated IDP conformational ensembles and dynamics.

## Methods

### Sample preparation
Recombinant p53-TAD (1–73) was expressed as a fusion protein with an N-terminal $Z_2$ domain using a modified pET28a vector[86]. Perdeuterated $^{15}$N-labelled p53-TAD (1–73) samples were expressed at 25 °C in *E. coli* adapted to 100% $D_2O$ minimal medium supplemented with $D_7$-glucose as the carbon source and $^{15}$N-NH$_4$Cl as the nitrogen source. Protein expression was induced with 1 mM IPTG. The expression culture was harvested 12 h after induction. Recombinant p53-TAD was purified using immobilised metal affinity chromatography on Ni-NTA resin (Macherey-Nagel, Germany) followed by Tobacco Etch Virus (TEV) protease cleavage at room temperature. The cleaved protein was reloaded onto Ni-NTA resin to remove the $Z_2$ domain and TEV protease. Gel filtration on a Superdex 75 16/60 HiLoad column (GE Healthcare) was performed to further purify the p53-TAD (1–73) fragment. Due to cloning, the WT fragment included an additional N-terminal Gly-Ser-extension and the P27A mutant a Gly-Ser-His-Met-extension. The fractions containing the purified protein were combined, concentrated with a 10 kDa concentrator (Vivascience).

### Sample conditions
Nuclear Magnetic Resonance (NMR) experiments were performed on a 1.5 mM p53-TAD sample in 50 mM sodium acetate buffer at pH 6.3, containing 50 mM sodium chloride and 0.03% sodium azide. Backbone amide $^1$H Off-resonance relaxation dispersion ($R_{1\rho}$-RD) experiments at temperatures ranging from 262 K to 265 K were carried out in glass capillary tubes to produce super-cooled conditions below the freezing point of water. Each capillary of 1 mm outer diameter (Wilmad, Buena, New Jersey) contained 25 µl of the p53-TAD sample, and 12 such capillaries were placed inside a 5 mm NMR sample tube. The sample was purposefully not labelled with $^{13}$C nuclei to avoid the necessity of $^{13}$C decoupling, which could be an extra source for RF heating. In addition, the heteronuclear J coupling of $C_\alpha$ and carbonyl carbon to the nearby amide proton and nitrogen nuclei can be a source of artifacts in relaxation dispersion profiles.

All NMR sub-$\tau_c$ relaxation measurements and $^1$H$_N$ extreme power Carr-Purcell-Meiboom-Gill (CPMG) RD measurements were performed on uniformly perdeuterated, $^{15}$N labelled proteins (both p53-TAD WT and P27A mutant) back exchanged with 100% $H_2O$. The samples were finally buffer exchanged to 50 mM sodium acetate buffer at pH 6.3 containing 50 mM NaCl, 5% (vol/vol) $D_2O$, and 0.02% sodium azide. The final p53-TAD protein concentrations of WT and P27A samples were 1.0 mM and 0.7 mM, respectively. Each sample was transferred to a 2 mm capillary and was placed within the magnet using a Bruker Match insert assembly.

### NMR relaxation measurements
All NMR spectra were collected with a Bruker Avance III HD spectrometer and corresponding Topspin 3.7 software operating at 950 MHz equipped with a TCI 5 mm cryo-probe and a Bruker Neo spectrometer (Topspin 4.0) operating at 1.2 GHz $^1$H field strengths, equipped with a TCI 3 mm cryo-probe. Sample temperature was controlled with dry $N_2$ gas using Bruker BCU-II VT units with medium chiller strength and 670 litre/hour gas flow for all experiments. Temperatures over the 263–298 K range were calibrated using a 3 mm Greisinger GMH 3750 thermometer equipped with a thermocouple. All spectra were referenced with respect to the water peak.

2D–$^{15}$N–$^1$H spectra, collected with a FAST-HSQC[87] pulse sequence at each temperature, were used to validate sample conditions. Assignments were transferred from previously published sources[88] at 298 K and propagated to spectra collected at 263 K by recording a series of 2D-HSQC spectra at 5 K temperature intervals. All 2D-NMR data were processed within the UNIX software environment NMRPipe[89] (version 12.6) and were further analysed and visualised using the software package nmrfam-sparky[90] (version 3.190).

The $^{15}$N $R_1$, $R_{1\rho}$ experiments were used for mapping points of the spectral density function[10] and recorded at 298 K, and 263 K under 950 MHz $^1$H field strength using standard protocols[91] and an eight-point measurement scheme (with two repeat points). $^{15}$N-$R_{1\rho}$ relaxation rates were measured using a spin lock field of 2 kHz[92]. Heteronuclear $^{15}$N nuclear Overhauser effect (NOE) measurements were recorded with 5 s mixing time using standard protocols[91] at 298 K under 950 MHz $^1$H field strength.

The amide backbone $^1$H Off-resonance $R_{1\rho}$ RD experiments were recorded for the WT p53-TAD at supercooled temperatures 262 K, 263 K, 264 K, and 265 K at a 950 MHz $^1$H field[93,94]. This narrow temperature range was limited by two constraints, (1) the requirement that the exchange process remains within the sensitivity window of NMR RD, and (2) sample freezing at 261 K even under the supercooled conditions inside capillaries. The pulse sequence of ref. 94 was taken. The spin-lock period $t_{rel}$ is flanked by adiabatic half-passage pulses (amplitude tailored according to the tanh function, frequency sweep controlled by the tan function, sweep width 100 kHz, duration 4 ms, the two half-passage pulses are related via time reversal)[95]. These pulses are integrated in the spin-lock period such that their maximum

amplitude is equal to the spin-lock amplitude (up to 35 kHz). The proton RF field is applied at water line, except during the spinlock where the carrier is positioned at 8 ppm±Δ, where Δ = √2 × spinlock amplitude. Nitrogen decoupling during the acquisition is achieved using a WURST sequence[96] with maximum RF field strength of 1.2 kHz. The phase cycle is: $\phi_1$ x,-x; $\phi_2$ 2x, 2y, 2(-x), 2(-y); $\phi_3$ and $\phi_4$ x; $\phi_{rec}$ x,-x, -x, x. Quadrature detection in F1 is achieved using the sensitivity enhancement scheme[97,98] by recording two datasets with ($\phi_4$, g5) and ($\phi_4$ + 180°, -g5) for each $t_1$ increment. Phase $\phi_1$ is incremented in concert with receiver phase to shift axial peaks to the edge of the spectrum[99]. Gradient pulses g1 to g7 are applied with durations 1.0, 0.5, 1.0, 1.0, 1.25, 0.5, and 0.125 ms and amplitudes 5.0, 4.0, 20.0, 12.0, 15.0, 4.0, and 14.86 G/cm, respectively. The delays are $\tau_a$ 2.25 ms, $\delta_1$ = 1.5 ms, $\delta_2$ = 0.5 ms. The spin-lock power was varied between 1 kHz and 35 kHz during the measurement to obtain effective transverse relaxation rates at different RF field. A 298 K timescale between 66 and 100 ns ( ± 1$\sigma$ uncertainty) was estimated by Arrhenius extrapolation from these four temperatures, validated against an Eyring estimate, and the uncertainty of this estimate was determined using Bayesian inference.

Site-specific $^{15}$N-transverse cross-correlated relaxation (CCR) rates ($\eta_{xy}$) were measured via 2D $^{15}$N TRACT (TROSY for rotational correlation times)[100,101] experiments performed at 298 K, and 263 K under 950 MHz and 1.2 GHz field strengths. The software NMRPipe was used to process all pseudo-3D spectra as well as to extract relaxation rates. Uncertainties in measured relaxation rates were estimated using error propagation from spectral RMS noise. Measured $\eta_{xy}$ values were converted[52,102] to approximate site-specific rotational correlation times ($\tau_c$) and further converted to approximate chemical exchange free intrinsic $^{15}$N and $^1$H transverse auto-relaxation rates ($R_{2,0}$) as published[103] using Python (version 3.8) scripts. For all calculations, standard values[52] for $^{15}$N–$^1$H bond length 1.02 Å, $\theta_{xy}$ = 17°, $^{15}$N CSA = -160 ppm, and $^1$H$_N$ CSA = 10 ppm[104] were used. The effect of varying these parameters in the calculation of $\tau_c$ has been described in detail elsewhere[52]. The relaxation data was visualised using OriginPRO software (version 10.1).

One set of $^1$H$_N$ Carr–Purcell–Meiboom–Gill (CPMG) experiments using extreme power were recorded at 263 K and 298 K with the published pseudo-4D IP/AP scheme pulse sequence[105] with modifications as published in ref. 45, required to run the experiment at extreme CPMG frequencies[106]. The experiments[106–109] were performed with the power on the $^1$H channel set to 18 W ($^1$H 90° pulse length ~ 8.1 μs) on the 950 MHz spectrometer and ~ 16.3 W ($^1$H 90° pulse length ~ 6.25 μs) on the 1.2 GHz spectrometer. A recycle delay of 3 s for all experiments was used to ensure minimal sample heating and a low-duty cycle. All experiments were recorded with 128 initial dummy scans to equilibrate the spin system and sample temperature before data acquisition. For the 1.2 GHz experiments, the constant CPMG duration ($T_{CP}$) was set to 20 or 40 ms, and 28 points were sampled in the CPMG frequency dimension (including the reference plane and two repeat points) ranging from 100 Hz up to 40 kHz. For each experiment, 100–120 (indirect dimension) and 1536 (direct dimension) complex points were recorded with 16 scans using 28 and 16 ppm spectral width along the indirect and direct dimension, respectively. The experiments were recorded with a 2 ppm bandwidth E-BURP refocusing pulse (centred at the middle of the $^1$H$_N$ region ~ 8.0 ppm) at the centre of the CPMG duration. The total experiment time was ~ 90 h at each temperature for each sample. The experiments at 950 MHz were acquired with 30 points (including the reference plane and two repeat points) in the CPMG frequencies dimension, ranging from 100 Hz up to 30.7 kHz, and with a hard pulse at the centre of the CPMG duration. For each experiment, 120 (indirect dimension) and 1024 (direct dimension) complex points were recorded with 4 scans, totalling ~ 24 h experimental time per sample at each temperature.

For all experiments, spectra were processed, and relaxation rates were extracted separately for the IP and AP sets of spectra using the UNIX software environment NMRPipe, followed by averaging for subsequent analysis. The differences between values from the IP and AP datasets were minimal. Site-specific solvent exchange contributions to measured $^1$H$_N$ $R_2$ values at 298 K were estimated at 950 MHz, using differences of site-specific $^1$H $R_1$ values acquired from two sets of inversion recovery pulse sequences recorded with standard parameters and recycle delays of >10 s. In the first experiment, water magnetisation was kept along the z-axis, whereas in the second experiment, it was completely dephased with a low gradient[110].

All fourteen proline NH resonances (Fig. 1A) were absent in all NH HSQC planes. In the relaxation measurements for the WT p53 sample at 263 K, the resonances of residues D7 and D57 overlapped, and at 298 K the resonances of residues W53 and F54 overlapped, both due to limited resolution in these experiments. Similarly, overlaps occurred for the P27A p53 mutant sample at 263 K for resonances of residues D7 and D57, and at 298 K for resonances of residues D7, L22, W53, W54, and D57. For these, relaxation rates could not be determined.

The 2D [$^1$H-$^1$H]-NOESY experiment was collected on a Bruker Avance III HD spectrometer operating at 900 MHz equipped with a TCI 5 mm cryo-probe at 298 K. The NOESY experiment was performed with 120 ms mixing time and 1024 and 512 complex points along $t_1$ and $t_2$ dimensions, respectively. The NOESY data were processed using NMRPipe[89] and analysed with nmrDraw (version 12.6) and CARA[111] (version 1.9.9).

## Molecular dynamics simulations

All simulations were performed using the molecular dynamics (MD) simulation software package GROMACS Version 2019.3[112]. Starting structures for each of the 30 trajectories were generated by first collapsing a fully extended p53-TAD (1–73) molecule performing a short generalised Born implicit solvent (GBSA) simulation[113]. From these trajectories structures were selected which were not fully collapsed (radius of gyration >3.5 nm) and did not contain any secondary structure elements. The p53-TAD (residues 1–73) WT protein and the P27A mutant were placed within dodecahedral boxes with an initial volume of 1222 nm³ (edge lengths: 12.0, 12.0, 8.485 nm) solvated in water and 100 mM NaCl. Virtual sites were used to allow for a time step of 4 fs[114]. The LINCS algorithm[115], applying a sixth-order iterative restraint on the bond distances, was used. The Particle Mesh Ewald (PME) algorithm[116] was used for electrostatic interactions with a cut-off of 1.0 nm. A reciprocal grid of 96 × 96 × 96 cells was used with fourth order B-spline interpolation. A single cut-off of 1.215 nm was used for the Van der Waals interactions. Neighbour searching was performed every 60 steps. Temperature was controlled by the velocity-rescale algorithm[117] with a 298 K target temperature and a 0.1 ps coupling constant; for pressure coupling the Parrinello-Rahman algorithm[118] was used with 1 bar target pressure and a coupling constant of 20.0 ps. All fully hydrated systems were allowed to equilibrate for 10 ns before the final production runs were started. Protein coordinates were recorded every 100 ps. All WT simulations were 20 μs long and all P27A simulations were 60 μs long to achieve better convergence for the expected slower P27A timescales. The Amber99sbws[119] force field was used in combination with the TIP4P2005s[120] water model.

## Exchange related relaxation: $R_{2,ex}$

Conformational dynamics such as folding and unfolding of an intrinsically disordered protein (IDP) induce changes in the chemical shift of the involved nuclei and were probed by CPMG RD experiments as described above. By fitting to the CPMG curve (Eq. (18)), the characteristic exchange timescale between the folded and unfolded states was determined for those residues for which the exchange rate falls within the detectable frequency range set by the maximum applicable RF power and where the chemical shift difference is sufficiently large. RD profiles were calculated from our MD simulations via the power

spectrum of the combined chemical shift time traces $CS_{HN}(t)$, calculated for all available trajectories of the respective nuclei. Chemical shifts of the p53-TAD backbone protons were predicted using the software SPARTA+[61] from simulation frames separated 100 ps in time from 20 µs trajectories for the WT and 60 µs trajectories for the P27A system. As shown by Xue et al. earlier, the RD profile is derived from the chemical shift autocorrelation functions[121,122], which we have calculated from $CS_{HN}(t)$ via the properly normalised power spectrum, i.e. the absolute-squared Fourier transform

$$R_{2,\,ex}(\nu) = \frac{1}{2} N \Delta t \left| \mathscr{F}\left(CS_{NH}(t)\right) \right|^2 \tag{1}$$

using the Wiener-Khinchin theorem[123]. Here, $\nu$ is the CPMG frequency, $N$ is the number of simulation frames, $\Delta t$ is the time step between frames (100 ps). Note that the unit of $CS_{HN}(t)$ is rad/s. RD profiles were calculated separately from each MD trajectory and then averaged.

### Tumbling related relaxation: $R_{2,0}$

To estimate $R_{2,0}$, the spectral density function (SDF) was calculated for each residue as described subsequently. From the SDF of each residue, $R_{2,0}$ was calculated via Eq. (13) and used as the offset for the calculated RD profiles.

### Calculation of the spectral density function and derived NMR observables $\eta_{xy}$, $\tau_c$, $R_1$, $R_2$, and NOEs

SDFs $J(\nu)$ were calculated for each residue from the Fourier transformation

$$J(\nu) = \frac{1}{5} t_{max} \mathrm{Re}(\mathscr{F}(C(\tau))), \tag{2}$$

of the backbone H-N-bond rotation-autocorrelation function $C(\tau)$[71,124]

$$C(\tau) = \langle P_2(\boldsymbol{\mu}(t) \cdot \boldsymbol{\mu}(t+\tau)) \rangle, \tag{3}$$

where $\tau$ is the lag time, $\boldsymbol{\mu}(t)$ is the unit vector of the covalent bond, $P_2$ is the second Legendre polynomial, $t_{max} = 100$ ns is the longest lag time considered, and $\langle \rangle$ denotes the time average along the trajectories. Due to the symmetry of the autocorrelation function, the real part of the Fourier transform is used for this calculation.

By combining the values of the SDF at specific frequencies, the transverse cross-correlation rate constant $\eta_{xy}$ and tumbling timescale $\tau_c$ were calculated using Eqs. (4)–(8), respectively, as described by Robson et al.[52],

$$\eta_{xy} = p \delta_N (4J(0) + 3J(\omega_N))(3\cos^2\theta - 1), \tag{4}$$

where

$$p = \frac{\mu_0 \gamma_H \gamma_N h}{16\pi^2 \sqrt{2} r^3}, \tag{5}$$

$$\delta_N = \frac{\gamma_N B_0 \Delta\delta_N}{3\sqrt{2}}, \tag{6}$$

$$J(\omega) = \frac{2\tau_c}{5[1 + (\tau_c \omega)^2]}, \tag{7}$$

and $\mu_0 = 1.27 \times 10^{-6}$ H m$^{-1}$, $\gamma_H = 267.52$ rad s$^{-1}$ T$^{-1}$, and $\gamma_N = -27.12$ rad s$^{-1}$ T$^{-1}$ are the gyromagnetic ratios of proton and nitrogen nuclei, respectively; $h = 6.63 \times 10^{-34}$ Js is Planck's constant, $\Delta\delta_N = 160$ ppm, $r_{NH} = 1.02$ Å, $B_0 = 28.19$ T, and $\theta = 17°$.

Similarly[52],

$$\tau_c = \frac{5c_1}{24} - \frac{336\omega_N^2 - 25c_1^2\omega_N^4}{24\omega_N^2 \left(1800c_1\omega_N^4 + 125c_1^3\omega_N^6 + 24\sqrt{3}\sqrt{21952\omega_N^6 - 3025c_1^2\omega_N^8 + 625c_1^4\omega_N^{10}}\right)^{\frac{1}{3}}} + \frac{\left(1800c_1\omega_N^4 + 125c_1^3\omega_N^6 + 24\sqrt{3}\sqrt{21952\omega_N^6 - 3025c_1^2\omega_N^8 + 625c_1^4\omega_N^{10}}\right)^{\frac{1}{3}}}{24\omega_N^2}, \tag{8}$$

where

$$c_1 = \frac{\eta_{xy}}{p\delta_N(3\cos^2\theta - 1)}. \tag{9}$$

Next, the rate constants $R_1$ and $R_2$ were calculated via Eqs. (10)–(15) as described by Palmer III.[24],

$$R_1 = \frac{d_2^2}{4} (3J(\omega_N) + J(\omega_m) + 6J(\omega_p)) + c_2^2 J(\omega_N), \tag{10}$$

where

$$c_2 = \frac{\omega_N \Delta\delta_N}{\sqrt{3}}, \tag{11}$$

$$d_2 = \frac{\mu_0 \gamma_H \gamma_N h}{8\pi^2 r^3}, \tag{12}$$

and $\omega_m = \omega_H - \omega_N$ and $\omega_p = \omega_H + \omega_N$, as well as

$$R_2 = \frac{d_3}{8} \left(4J(0) + 3J(\omega_H) + J(\omega_m) + 6J(\omega_N) + 6J(\omega_p)\right) + \frac{1}{6} c_3 \omega_H^2 \left(4J(0) + 3J(\omega_H)\right), \tag{13}$$

where

$$c_3 = \frac{\Delta\delta_H^2}{3}, \tag{14}$$

$$d_3 = \left(\frac{\mu_0}{4\pi}\right)^2 \left(\frac{h}{2\pi}\right)^2 \gamma_H^2 \gamma_N^2 r^{-6}, \tag{15}$$

and $\Delta\delta_H = 10^{-5}$ is the difference in the axially symmetric proton chemical shift tensor.

Finally, NOEs were calculated as described by Farrow et al.[72],

$$\mathrm{NOE} = 1 + \frac{d_3^2 \gamma_H}{4\gamma_N} \left(6J(\omega_p) - J(\omega_m)\right) \frac{1}{R_1}, \tag{16}$$

where $d_3$ is defined in equation (15) and $R_1$ in Eq. (10).

The code for the above calculations, as well as sample calculations and sample data, were deposited at the public repository GitHub https://github.com/dszollosi/p53_TAD_dynamics

### Fitting of 2-state CPMG model to RD profiles measured by NMR at 263 K

Characteristic timescales $\tau$ and their uncertainties were derived from NMR CPMG measurements at 263 K via Bayesian inference. The posterior distribution was estimated by Monte Carlo sampling (4000 samples) using the pymc5 package[125] (version 5.15.0) with Python 3.11.11. Posterior probabilities were calculated from the probabilities of the set of RD measurements for CPMG profiles described by

Eq. (17) using a Gaussian experimental error distribution, the width σ of which was also subject to inference. For some of the residues of the P27A mutant (e.g., Phe19 or Asp21, see Supplementary Information Fig. 1), the RD profile indicated the presence of two relaxation processes with markedly different timescales. For these residues, a sum of two CPMG profiles with a single $R_{2,0}$ offset was estimated, Eq. (18). For σ and $R_{2,0}$, a uniform prior with limits from 0 to the maximum of the RD profiles was used. For $\Phi$ and $\tau$, log-uniform priors with boundaries of $10^1$–$10^9$ 1/s$^2$ and $10^{-8}$–$10^{-4.5}$ s, respectively were used; for profiles that indicated a second process the additional $\Phi$ parameter used the same boundaries while $\tau_2$ was searched between $10^{-4.5}$ and $10^{-3}$ s. Estimated experimental errors (see https://doi.org/10.17617/3.JWVPWJ) were found to be similar in magnitude to the posterior distribution for σ.

$$R_{2,\text{eff}}(\nu) = R_{2,0} + \phi\tau\left(1 - 4\nu\tau\,\tanh\frac{1}{4\nu\tau}\right), \tag{17}$$

$$R_{2,\text{eff}}(\nu) = R_{2,0} + \phi_1\tau_1\left(1 - 4\nu\tau_1\,\tanh\frac{1}{4\nu\tau_1}\right) + \phi_2\tau_2\left(1 - 4\nu\tau_2\,\tanh\frac{1}{4\nu\tau_2}\right). \tag{18}$$

### Fitting of 2-state CPMG model and stretched CPMG model to RD profiles calculated from MD simulations

To compare our NMR CPMG measurements with our MD simulations not only in terms of the raw spectra (see above), but also in terms of the characteristic exchange dynamics timescales $\tau$, the analytical form of CPMG profiles for a two-state model (Eq. (19))[55,126]

$$R_{2,\text{MD}}(\nu) = \phi\tau\left(1 - 4\nu\tau\,\tanh\frac{1}{4\nu\tau}\right), \tag{19}$$

was fitted both to the measured NMR RD spectra as well as to those calculated from our MD simulations. Here, $\nu$ is the CPMG frequency, $\Phi = p_A\, p_B\, \Delta CS^2$ is the population weighted chemical shift variance, $p_A$ and $p_B$ are the populations of the two states A and B of the model, respectively, and $\Delta CS$ is the chemical shift difference between these two states.

Inspired by the shape of the RD profiles calculated from the simulations (cf. Supplementary Information Fig. 4), which in the logarithmic plot appears to be 'stretched' in the frequency domain relative to the above CPMG two-state model, a generalised model in terms of a 'stretched' function was considered,

$$R_{2,\text{MD}}(\nu) = \phi\tau\left(1 - (4\nu\tau)^\gamma\,\tanh\frac{1}{(4\nu\tau)^\gamma}\right), \tag{20}$$

similar in spirit to the 'stretched exponential functions' used by Frauenfelder to describe the multi-tier dynamics of folded proteins[5,66,127]. Salvi et al.[67] used a weighted superposition of $R_2$ transverse relaxation rates with discrete timescale (fast, intermediate, slow). This is an alternative approach to include a broad range of characteristic timescale which we here described by the additional fitting parameter $\gamma$ in Eq. (20). Accordingly, the stretched CPMG profile can be interpreted as a weighted superposition of many two-state CPMG profiles with a broad range of characteristic timescales, which is described by the additional fitting parameter $\gamma$. Therefore, $\gamma$ also characterises the distribution width of barrier heights governing p53-TAD dynamics, namely $\gamma = 1$ indicates a two-state process and the smaller the value the broader the timescale distribution is. Fits of Eqs. (19) and (20) to either measured or calculated RD profiles were performed using a Bayesian approach[128] using the pymc5 Python package[125], with $\tau$ and $\Phi$ (and, additionally, $\gamma$ for Eq. (20)) as free parameters to be determined by the fits.

The fact that all RD profiles were calculated by averaging over 30 individual RD profiles, each calculated from an absolute-squared

Fourier transform of chemical shift trajectories as described above, required particular attention. Notably, Fourier transforms calculated numerically from finite time series are notoriously noisy, and the distribution of each individual (absolute squared) Fourier coefficient $R_k := R_2(\nu_k)$ for realisations with a uniform distribution of random phases follows an exponential function[129],

$$p(R_k) = \frac{1}{R_k^0}e^{-\frac{R_k}{R_k^0}}, \tag{21}$$

where $p(R_k^0)$ is the true coefficient from which the realisations were drawn. Hence, the probability distribution of the mean $\bar{R}_k$ of $N = 30$ Fourier transforms of realisations of the same process (the phases of which are assumed to be statistically independent and uniformly distributed) follows a gamma distribution

$$p(\bar{R}_k) = \frac{\beta^N \bar{R}_k^{N-1} e^{-\beta\bar{R}_k}}{\Gamma(N)}, \tag{22}$$

where $\beta = \frac{N}{R_k^0}$.

Accordingly, this probability distribution was used (instead of a Gaussian distribution) for the Bayesian inference of each Fourier coefficient, with the fitting target $\bar{R}_k$. A log-uniform prior distribution between $10^3$–$10^9$ Hz$^2$ was used for $\Phi$, a log-uniform prior between 1 ns–100 μs for $\tau$, and a uniform prior between 0–2 for $\gamma$. An additional Jupyter notebook describes in detail all fitting steps, available at the public repository GitHub https://github.com/dszollosi/p53_TAD_dynamics.

For the residues 49ASP, 54PHE, 56GLU, 57ASP, 62GLU, 63ALA, and 66MET, the RD profile does not reach a plateau at low frequencies, indicating very slow dynamics (cf. the respective panels in Supplementary Information Figs. 4 and 5) which are also apparent from the Bayes posteriors. For these residues, therefore, no relaxation timescales were derived from the fits, which are therefore also not shown in Fig. 2B.

For an illustrative example comparing the different fitting models described above, see Supplementary Information Fig. 6 and the respective Supplementary Information Section.

### Markov state model analysis

The structural dynamics of the p53-TAD WT helix 1 and helix 2 comprising residues 18–26 and 40–53 respectively, were analysed by building a Markov state model from a dimension-reduced representation of the 600,000 available structure snapshots using the Python library "Deeptime"[130] version 0.4.4. To extract the Markov states from the MD simulations, clustering was performed on the combined 600 μs trajectory set using the WT backbone dihedral angles $\varphi$ and $\psi$ of all helix 1 residues and snapshots from the MD trajectories recorded every nanosecond. For the clustering, sine and cosine values of these angles were used to avoid periodicity discontinuities, such that each residue is described/characterised by $\sin(\varphi)$, $\cos(\varphi)$, $\sin(\psi)$, and $\cos(\psi)$. Using these internal coordinates, super-positioning of snapshots of this IDP was avoided, which would have been challenging.

Dimension reduction of the resulting structure vectors was performed using time-lagged independent component analysis (TICA)[65], with a lag time of 10 ns. The first three independent components were used as input for subsequent KMeans clustering[131] (using the Deeptime implementation requesting 150 clusters, init_strategy was 'kmeans++', max_iter = 500, and fixed_seed = 13). This clustering served to assign all simulation frames to microstates, which served as input for building a hidden Markov model requesting 15 initial macrostates. These 15 initial macrostates were merged heuristically into a final number of seven macrostates, chosen empirically by inspecting spatial proximity of

**Table 1 | Combining initial into final Markov states**

| Initial state | Final state |
|---|---|
| 13 | 1 |
| 15 | 2 |
| 8 | 3 |
| 7, 9, 12 | 4 |
| 10, 14 | 5 |
| 1, 2, 4, 6 | 6 |
| 3, 5, 11 | 7 |

A total of 15 initial macrostates (left column) were combined into seven final macrostates (right column) for p53-TAD WT helix 1.

average structures in the TICA projection and also considering the number of α-helix stabilising hydrogen bonds, to extract structurally and chemically unique and distinct conformations (Table 1). Hydrogen bonds stabilising an α-helix or a $3_{10}$-helix were defined by donor-acceptor distance $d$ (in Å) according to the empirical Espinosa hydrogen bonds energy estimate[132] (in kJ/mol) $E_{HB} = -25,300e^{-3.6d}$ with a cut-off hydrogen bond strength of $1\,k_{B}T$. As a result, hydrogen bonds were counted for all donor-acceptor pairs closer than $d = 0.256$ nm.

### Identification of metastable tertiary structure elements

p53-TAD transiently forms tertiary structures more complex than single α-helices or β-sheets. To detect those tertiary structures that last longer than 1 μs and therefore contribute to the overall structural ensemble, we performed a $C_{\alpha}$-$C_{\alpha}$ distance fluctuation analysis on our MD trajectories (both for the WT and for the P27A mutant), which uses internal coordinates and therefore does not require orientational fitting. Specifically, for every trajectory frame, intramolecular distances between all pairs of $C_{\alpha}$ atoms were calculated. Residue pairs separated by no or only one residue along the sequence were excluded from this analysis, because their mutual $C_{\alpha}$-$C_{\alpha}$ distances showed only small fluctuations and thus do not provide much information on metastable tertiary structures.

For all trajectories, the size of the distance fluctuations was quantified over time and for each residue pair via standard deviations, calculated by averaging over a sliding window of 100 ns width (containing 1000 frames). For each window position in time, and for each residue, the standard deviation of the distances to all other residues was calculated and then sorted. From the obtained list of standard deviations, the 10th smallest one, indicating the 10th least fluctuating distance, was used as a tertiary structure indicator. Here, the 10th distance was chosen empirically, to best reflect tertiary structures larger than a typical two-turn α-helix or a smaller β-sheet. Of note, this indicator was also chosen because it does not require the involved residues to be on a continuous sequence segment and, therefore, it also identifies folds formed by distant residues.

Figure 4A shows an example of the resulting fluctuation map, showing averaged distance fluctuations between 0.1 nm (yellow) and 0.5 nm (dark blue), plotted for every residue (y-axis) over time (x-axis). In this plot, regions extending vertically over several residues and horizontally over more than 1 μs are easily identified, e.g., the two large yellow regions involving residues 15–53 lasting from 2–6 μs, and the one involving residues 47–66 between 20 and 24 μs. This analysis also recovers the faster tier 1 folding dynamics of the two helices of p53-TAD, which are sufficiently stable to show up as rapidly fluctuating yellow bands between residues 18–26 (helix 1) and residues 40–53 (helix 2). For an automated scan of all MD trajectories, a fluctuation cutoff of 0.15 nm was chosen, and 93 transient tertiary structure elements were identified lasting longer than 1 μs.

One might expect the transient tertiary structures indicated by the MD simulations and tier 0 dynamics to be detected on a NOESY spectrum. The fact that not one but many different transient tertiary

structures are seen also explains why these are not seen as peaks in the NOESY spectra. Indeed, inspection of the expected positions in measured NOESY spectra (see "Methods" subsection "NMR relaxation measurements") did not reveal any signals beyond the noise level. Although the population of tertiary structures, taken together, is large enough to evoke a low-frequency component in the RD profiles at least for P27A, the population of each of the many different structures is below 0.3%, whereas an estimated population of at least 1% would be required to generate a visible NOESY cross-peak.

### Analyses and comparison to polymer models

The extensive sampling provided by our atomistic simulations of the p53-TAD also allowed us to characterise the local fast (tier 2) dynamics of this IDP from a polymer model perspective, e.g., in terms of persistent lengths and end-to-end distances, which for a polymer chain— and depending on the polymer model used—are connected[133]. For, e.g., a wormlike chain model, the (average squared) end-to-end distance $R_{e}$ reads

$$\langle R_e^2 \rangle = 2l_p L \left(1 - \frac{l_p}{L}\left(1 - e^{-\frac{L}{l_p}}\right)\right), \qquad (23)$$

where $l_p$ is the persistence length, $L = bN$ is the contour length, i.e., the length of one polymer unit ($b = 0.38$ nm) multiplied by their number $N$. From our MD trajectories, $\langle R_e^2 \rangle$ was calculated by time- and trajectory-averaging over all distances between all pairs of $C_{\alpha}$ atoms separated by given contour length (i.e., number of residues). Fitting Eq. (23) to the resulting curve (Supplementary Information Fig. 19A) yielded a persistence length of $l_p = 1.0$ nm.

Residue-specific persistence lengths were obtained similarly, except that averaging over different sequence positions was omitted, such that $\langle R_e^2 \rangle$ was obtained as a function of 'start' residue. Similar fits to Eq. (23) as above provided residue-position-resolved persistence length, characterising deviations from a simple homopolymer (Supplementary Information Fig. 19B). As can be seen, shorter persistence lengths $l_p$ were obtained for proline rich segments, e.g. Pro12-Pro13.

We also characterised the relation between persistence length $l_p$ and radius of gyration $R_g$, which is given by the polymer scaling law[77]

$$\langle R_g \rangle = \sqrt{\frac{2l_p b}{(2\nu+1)(2\nu+2)}} N^{\nu}, \qquad (24)$$

where $N$ is the number of residues in the different tested segments, $l_p$ and $b$ are as defined above, and $\nu$ is the scaling exponent. A value of $\nu = 0.5$ indicates a 'Flory random coil'[134], smaller values a compact and larger values a more extended ensemble. Similarly, as for $\langle R_e^2 \rangle$, $R_g$ was calculated for all protein segments of length between $N = 6$ and 30 residues, averaging over time, trajectories, and all possible segment positions in the peptide. From the fit shown in Supplementary Information Fig. 19C a markedly shorter persistence length of 0.31 nm and a scaling exponent of 0.66 was obtained; the latter indicating an expanded coil state[77], similar to an excluded volume chain model (0.588)[135]. For comparison, $R_g$ of the full protein was calculated similarly, but was not used for the above fit.

Alternatively, the persistence length was also estimated (Supplementary Information Fig. 19D) by fitting single and double exponential functions, respectively, to the normalised orientation autocorrelation function of the vectors connecting $C_{\alpha}$ atoms separated by $N > 2$ residues[133],

$$C(N) = Ae^{-\frac{Nb}{l_{p,1}}} + (1-A)e^{-\frac{Nb}{l_{p,2}}}, \qquad (25)$$

where $A$ is a prefactor, $N$ is the number of polymer units (residues) separating the vectors, with the same unit length $b = 0.38$ nm as above. Persistence length ($l_p$) from the single exponential fit is $1.19 \pm 0.01$ nm,

whereas from the double exponential $0.84 \pm 0.01$ nm and $5.06 \pm 0.26$ nm, for $l_{p,1}$ and $l_{p,2}$, respectively.

The fast reorientation tier 2 dynamics were also characterised by the running average root mean squared deviation (RMSD)

$$\text{RMSD}(\tau) = \sum_{i=1}^{N_{\text{atoms}}} \sqrt{\frac{1}{N_{\text{atoms}}} \left( x_i(t) - x_i(t+\tau) \right)^2}, \qquad (26)$$

averaged over all trajectories (Supplementary Information Fig. 19E). With increasing lag time, and for ca. 10 ns over nearly two timescale decades, the structural deviation increase closely follows a power law with an exponent of 0.39; thereafter the RMSD saturates. This reorganisation time of ca 10 ns agrees with that measured for other polymers by single-molecule FRET[15].

Complementing this analysis of structural dynamics at a rather detailed level, we finally calculated the autocorrelation times of the radius of gyration $R_g$ as well as of the end-to-end distance of the full p53-TAD chain. Fitting a double exponential function

$$C(\tau) = Ae^{-\frac{\tau}{\tau_1}} + (1-A)e^{-\frac{\tau}{\tau_2}} \qquad (27)$$

to the respective autocorrelation functions that were calculated similarly as above, two timescale values ($\tau_1$ and $\tau_2$) were obtained for each of the two observables (Supplementary Information Fig. 19F), namely 22.0 and 406.8 ns for $R_g$ and 18.5 and 349.8 ns for the end-to-end distance.

### Reporting summary

Further information on research design is available in the Nature Portfolio Reporting Summary linked to this article.

## Data availability

Supplementary Movies 1–3 have been deposited at https://github.com/dszollosi/p53_TAD_dynamics [https://doi.org/10.5281/zenodo.19704215][136]. All NMR spectra have been deposited at https://edmond.mpg.de/dataset.xhtml?persistentId=doi:10.17617/3.JWVPWJ, https://doi.org/10.17617/3.JWVPWJ. All molecular dynamics simulation starting structures, simulation parameters, and the force field were deposited to the same repository, as well as coarse-sampled trajectories. The full molecular dynamics simulation trajectories are available from H.G. upon request; requests will be answered within 1 month. These data have not been deposited due to large data size (> 1 TByte). The following Biological Magnetic Resonance Bank entry was used: BMRB 17760 (backbone resonance assignments of p53 N-terminal transactivation domain (1–93)). The following Protein Data Bank entries were used: PDB 2ESX (V3 region of gp120 of the JR-FL HIV-1 strain); PDB 2BTB (NMR STUDY OF N-TERMINAL HUMAN BAND 3 PEPTIDE, RESIDUES 1–15); PDB 8Y3S (Human Keratin 19 head domain segment G28–G38 in solution); PDB 5MWP (The structure of MR in complex with AZD9977). Source Data have been deposited at https://edmond.mpg.de/dataset.xhtml?persistentId=doi:10.17617/3.JWVPW.

## Code availability

The program code used for all calculations, additional sample calculations, and sample data has been deposited at https://github.com/dszollosi/p53_TAD_dynamics [https://doi.org/10.5281/zenodo.19704215][136].

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

## Acknowledgements

We thank Claudia Schwiegk for protein sample production; Lars Bock, Eliane Briand, Nicolai Kozlowski, Bert de Groot, Florian Leidner, and Kyou-Hoon Han for discussions and suggestions; and Petra Kellers for help editing the manuscript.

## Author contributions

Conceptualisation: D.S., S.R., H.G., C.G., and S.P. Formal analysis: D.S., S.R., H.G., S.P., and D.M. Investigation: D.S., G.N., S.R., R.K., S.P., D.M., G.J.R., N.E., A.K.R., D.L., S.B., and M.H. Methodology: D.S., S.R., H.G., S.P., D.M., G.J.R., A.K.R., D.L., S.B., and M.H. Project administration: H.G. and C.G. Resources: H.G., C.G., and S.B. Software: D.S. Supervision: H.G. and C.G. Writing—original draft: D.S. and H.G. Writing—review & editing: D.S., S.R., H.G., C.G., D.M., S.P., and M.H.

## Funding

G.N. and S.R. were supported by the Alexander von Humboldt Foundation. Computer time by the Max Planck Compute and Data Facility, financial support by the DFG (project EXC 2067/1-390729940), the KBSI internal research programs (C539200, C523400, C526112, and C539110), and the Max Planck Society are gratefully acknowledged. Open Access funding enabled and organized by Projekt DEAL.

## Competing interests

The authors declare no competing interests.
