## [Transparent Peer Review file · Nature Communications]

Hierarchical multi-timescale structural dynamics of the disordered N-terminal of p53

Corresponding Author: Dr Helmut Grubmüller

Version 0:

Reviewer comments:

Reviewer #1

(Remarks to the Author)

The manuscript by Szöllősi et al. deals with dynamics of intrinsically disordered proteins (IDPs) across different time scales – a emerging topic of great interest and biological importance. The authors employ and combine advanced NMR spectroscopic techniques and molecular dynamics simulations to access time scales that have previously been difficult to access. For this, they make use of high power ^1H relaxation dispersion experiments that they have previously developed, combined with more classical NMR approaches, such as spin relaxation. In combination they thus experimentally tackle time scales in the picosecond to nanosecond time scale and in the low microsecond to millisecond time scale (and by extrapolation down to the hundreds of nanosecond time scale). Thereby, they decrease the gap of NMR accessible time scales, in which the IDP p53 shows dynamic features. Time scales faster than few microseconds are accessible by long MD simulations and compared with the experimental data. Their combined model reveals hierarchical dynamics of chain motion, and transient folding of secondary and small tertiary structures that the authors think may be relevant for function. They assume that other IDPs will show similar functional time scales.

The study presented touches a highly important topic and presents sound data that suggest the presence of dynamics that has not previously been uncovered. However, the manuscript is hard to read, since it is composed of a very short main text with an extensive supplementary file and an additional extended data file that cross-reference each other. Likely due to this partially confusing structure, the conclusions and what exactly has been done is not always clear, and this needs to be revised along the following points:

- In their introduction, the authors compare dynamic time scales that they address by NMR spectroscopy with those that have previously been addressed by single molecule spectroscopy. They mention time scales in the nanosecond range (line 68) and that the time scales between 100 nanoseconds and 10 microseconds remain elusive. This is only partially true, approaches based on fluorescence correlation spectroscopy, which the authors also cite, are sensitive from the picosecond time scale down to the millisecond time scale without time scale interruptions. In fact, classical interconversion times that have been found in IDPs using FRET and FCS combinations are often in the range around 100 nanoseconds. This should be adapted in the text accordingly.
- Related to this point, lines 103 to 111 require a more detailed discussion. IDP dynamics measured by single molecule fluorescence and NMR approaches rely on very different observables (e.g. distance fluctuations vs fluctuations in chemical shifts, which is much more local). This aspect needs to be discussed, because it affects the actual time scales expected and how they are interpreted.
- In Fig. S2, the authors show a comparison of ΔR_2 , which is the effective R_2 at the highest available CPMG frequency from which the R_2 calculated from TRACT experiments was subtracted. The figure shows that between the time scales accessible using high power CPMG experiments and the 'molecular tumbling time scale' further dynamics exist. The authors call this an 'otherwise blind spot'. However, also with their new experiments, they cannot access these time scales quantitatively. They should be more explicit about that. Furthermore, it is worth including the presence of these dynamics in some way into Figure 1G.
- The authors claim that they can assess protein dynamics down to 80 ns (tier 1) using RD. However, this is only by extrapolation from very cold temperatures. The authors should be a bit more clear on this. Furthermore, the extrapolation is based on experiments conducted at four different temperatures, from 262-265 K. These are very small differences compared to an exponential extrapolation to 298 K. The authors should discuss this aspect and the validity of their approximation.
- It remains unclear how the comparison of MD derived RD curves with the experimental data is performed. It seems that the

experimental data plotted in Fig. 2E-H and Fig. S4 should be the same as those in Fig. 1C-E and Extended Data Fig. 1. However, this is apparently not the case. Both types of plots have at their y-axis R_{eff} (1/s), but the values of the experimental data points are dramatically different between Figs. 2E-H/S4 and Figs. 1C-E/Extended Data Fig. 1, and so are the shapes of the curves. Have the authors performed some operation with the data before comparing with the MD derived RD curves? Maybe this information got lost somewhere in between the three files (maintext, extended data and supplement). However, understanding this aspect is crucial for judging the implication of the comparison of time scales between NMR and MD. A clear explanation for how the data were treated is thus necessary to understand this aspect.

- Related to the above point, the authors also mention some 'high-power artifacts' that increased the measured RD profiles at higher frequencies. Can the authors be more clear what artifacts they are talking about?
- Also related to the same point is the interpretation of Fig. 2B, which shows $R_{2,eff}$ at a CPMG frequency of 100 s⁻¹. The values agree with those plotted in Figs. 2E-H and Fig. S4, but not at with those plotted in Figs. 2E-H/S4 and Figs. 1C-E/Extended Data Fig. 1. The authors should be very clear about what exactly they are visualizing.
- In the Supplementary Information, the authors say that they compare MD derived orientational correlation times with the TRACT relaxation times and cite Fig. 2B. However, Fig. 2B apparently shows $R_{2,eff}$ at a CPMG frequency of 100 s⁻¹ (see above). Which is correct?
- In the MD data analysis, the authors identify different Markov states of which they determine the interconversion. They start out with a certain number (15 for the wild type), which they then cluster further into 8. Can the authors be more explicit on how these states were defined given their proximity in the TICA space? What were the criteria for reducing 15 states into 8? And how does the initial choice of number of states affect the results?
- The comparison between MD simulations and FRET distance distributions needs clarification: It is not clear, what equations the authors used to calculate distances from FRET efficiencies. Was the IDP nature (rapid sampling of distances) taken into account for example? How were the dyes (sometimes fluorescent proteins appear to have been used, which are quite big) considered in the comparison? The authors also talk about experimental and MD derived distance distributions, but only show a single distance for the experimental data. Can they clarify this? It would be helpful to transparently lay out the experimental conditions of every experiment that has been compared and explain the conditions under which the distances have been compared, respectively.
- The authors point out the hierarchical time scales they find in an IDP and a 'folding funnel' similar to folded proteins (line 235). In fact, and this is what also their comparison of long lived structural elements with the PDB shows, the authors are identifying small (transiently) structured elements within the IDP – and these are subject to hierarchical time scale motions. This seems not to be the case for fully disordered regions with no secondary structure. This also goes along with the notion that a continuum exists between disordered and folded proteins. The authors should discuss this aspect.
- The above point also relates to the discussion section, where the authors describe that their work 'challenges the prevailing view that the lack of stable folded structures of IDPs implies an unstructured and shallow free energy landscape that only gives rise to [...] tier 2 [...] dynamics'. It is now accepted that a continuum of folded and disordered structures exist and that many IDPs possess transiently folded elements. This view does not take away any of the novelty of the presented manuscript, since slower timescale motion have indeed not often been observed in unbound IDPs by NMR spectroscopy and certainly not on the extended time scale presented here. However, this aspect has to be discussed in a balanced manner. Furthermore, citation of Nettels et al. at this position seems somewhat misplaced. In that article, dynamics on the order of 40-70 ns have been found, which might much rather correspond to tier 2 (or in between tier 1 and tier 2). In addition, the article deals with a guanidinium hydrochloride unfolded protein, which will certainly sample dynamics associated with transient structures to a much lesser extent than an actual IDP. The authors should consider this in their revision.
- The idea that Measles Virus N-TAIL possesses similar hierarchical dynamics as p53 is appealing. However, experimental data would be needed to support this claim – especially since the authors point out that the choice of the force field is strongly dependent on the protein under study.

Minor points:

- The P27A mutant is only mentioned somewhat on the side. A dedicated paragraph to introduce this mutant and what is expected from it would benefit the readability of the manuscript.
- The spectral density function is sometimes referred to as 'SDF' and sometimes as $J(\nu)$. A uniform expression should be adapted throughout the manuscript.
- The authors state that 'contributing to the NMR RD profiles are also faster, ns and ps motions...'. This sentence merits another phrase to describe how these change the relaxation dispersion curves.
- In the context of the analysis of $J(\nu)$, the authors state that there is again evidence for 'multi-state dynamics involving a wide spectrum of timescales also within the fast tier 2 regime' (lines 314-315). This agrees with dynamics that have been observed for other IDPs and should be supported by a small discussion along these lines.
- The statement that 'such transient structures are not merely biochemical curiosities' is maybe a bit too general, given that the authors have studied one protein. Maybe they can tune this expression down a bit.
- The authors mention 1H off-resonance $R_{1\rho}$ data in the methods section. It is unclear, where these data can be found in the manuscript.
- The authors should add a proper figure legend to Fig. S4.
- Figure S7: Is there any particular reason, why the authors compare wild type experimental data with P27A simulated data?
- Figure S8: Can the authors comment, why the R_2 is not reproduced so well and how this may affect the interpretation of their data? How this related to the other data comparisons undertaken? The green lines in the figure should also be labeled with the force field used under these conditions.
- Figure S9: It is interesting to see that the MD derived RD profiles all level off at an $R_{2,eff}$ of 0. This is not the case in Fig. 2E-H and Fig. S3. Can the authors comment on this?

- Extended Data Figure 2 shows chemical shift comparisons between MD derived and experimental chemical shifts. How do these plots look like when secondary chemical shifts are compared?
- Extended Data Figure 3: What are the horizontal lines in the experimental radii?
- Extended Data Figure 5: Considering the comparison of MD with PET-FCS measurements, it would be interesting to see until which time point exactly experimental data have been obtained, and compare this time point with when the black and cyan line start deviating from each other.
- Extended Data Figure 8: Can the authors explain the difference between the mutant NMR and MD data?
- The authors indicate supplementary movies. Where can they be found?

Reviewer #2

(Remarks to the Author)

This manuscript from the Griesinger and Grubmuller labs describes a recent collaborative effort to decode the dynamics of transient structure formation in the well-studied p53 N-terminal transactivation domain (TAD). Using high-powered relaxation dispersion methods and extensive molecular dynamics simulations, the authors obtain new insights into the identity of transiently formed secondary and tertiary structures in the p53 TAD and the timescales by which these structural elements interconvert. The experiments and analysis described in this work are rigorous and thorough and there is little to criticize about the technical aspects of this manuscript.

My enthusiasm for the manuscript overall is unfortunately dampened by the presentation of the data and the brevity of the text. As currently written and organized, the authors have failed to present the data in a manner that allows them to highlight the exciting potential of the state-of-the-art methods they have used here. In general, I struggled to understand why so much information was relegated to the Extended Data and Supplementary Information. Much of the text included in the Supplementary Information is absolutely essential for the reader to be able to interpret any of the results presented in the main text. I hope that the authors will make significant efforts to rewrite this manuscript in a way that thoroughly explains why certain experimental and computational methods were used, how the results from these experiments can be analyzed, and what sort of interpretation is possible. In its current iteration, the impact of the work is totally lost.

Some specific comments:

1. When the authors present the results of the relaxation dispersion experiments, they do not explicitly state why some data are not shown. This is particularly notable in Fig 1E where there is not a green curve shown despite the legend stating that there should be, but this is pervasive throughout. Presumably data are not shown for residues that either did not show dispersion or for residues that were not assigned and/or did not have detectable resonances in the spectra. This should be explicitly stated and explained.
2. Fig 1G – the WT data points for the Tier 2 dynamics are hard to decipher as they are largely hidden by the data for the P27A mutant.
3. The color scheme in Fig 2 is problematic. It is difficult to distinguish between the light and dark green in these figures. Please consider using colors with higher contrast.
4. The discussion of the stretched CPMG model in the main text is inadequate. What is the reader supposed to glean from the stretching factor and why this model would be appropriate to describe the data?

Reviewer #3

(Remarks to the Author)

Szöllösi, Pratihari et al. report a combined NMR and MD study of the p53-TAD, an intrinsically disordered region (IDR). In particular, they focus on characterizing the different timescales at play in the system as observed by relaxation dispersion NMR and modeled by MD. They find that this IDR is characterized by a complex hierarchy of timescales ranging from sub-nanoseconds to tens of microseconds, associated with structural rearrangements spanning from side-chain motions to tertiary structural fluctuations. They further suggest that this hierarchy of timescales resembles that of natively folded proteins and may underlie the ability of this region to interact with multiple partners.

The paper provides an interesting and detailed picture of the structural dynamics of p53-TAD; however, some of the claims appear too strong given the current data.

* Most of the timescales are measured under supercooled conditions to access fast dynamics and then extrapolated. It is not entirely clear whether the slower timescales were also probed at temperatures closer to physiological conditions.

* The slow ("tier-0") timescale is observed only for the P27A mutant. Its interpretation and extrapolation to the wild type through MD simulations are used to discuss the system as a whole. This is certainly an interesting observation, but claiming a hierarchy spanning seven orders of magnitude as a general property seems overstated given the current evidence (low temperature and mutation dependence).

* Following up on the previous point, the simulations do not reproduce the experimentally observed ~215 μ s slow exchange process in the mutant, instead identifying tertiary elements persisting only up to ~5 μ s. It is therefore not accurate to state that the simulations "accurately reproduce" the experimental hierarchy. A stronger validation would result from hypothesizing and testing a mutation derived from the conformations identified by the simulations.

* A χ^2 value quantifying the agreement between simulations and SAXS data is not provided.

* The PRE comparison is performed using C α -C α distances, which is a rather low-resolution approach given the availability of more accurate methods such as DEER-PREdict (10.1371/journal.pcbi.1008551).

* The Markov state model lacks analysis of implied timescales and other tests that would demonstrate the robustness of the

model.

* SAXS, FRET, and PRE data should be made available. While the trajectories are extensive, suitably pruned versions should also be shared for reproducibility.

In terms of broader context, slow conformational dynamics on the tens-of-microseconds timescale have previously been suggested using MD simulations for A β 42 (10.1038/s43588-020-00003-w), and a similar picture emerged in terms of free-energy barriers for A β 40 (10.1038/srep15449).

In conclusion, this is an interesting and valuable study providing important insights into the conformational dynamics of an intrinsically disordered protein. However, some claims should be moderated to avoid unsubstantiated overstatements, without detracting from the significance of the work. A validated prediction related to the tier-0 state would dramatically strengthen the manuscript.

Version 1:

Reviewer comments:

Reviewer #1

(Remarks to the Author)

The authors have made a tremendous effort to simplify the flow of the manuscript and therefore significantly strengthened it. The authors stress that different hierarchical time scales exist also in IDPs. While this is interesting, it is important, particularly in the last paragraph of the discussion section, to make sure to distinguish dynamics sampled by IDPs from those of folded proteins. The figures in the supplementary file are not cited in order, maybe this could still be adapted. In addition to this, only few comments remain to be addressed.

- Comparison of the different time scales in the results sections (lines 119-130) still appears somewhat incomplete. Both single molecule FRET as well as NMR are sensitive to fast nanosecond motions, which are much faster than the measured supra- τ_{auc} motions. While similar motions have been measured using PET-FCS, as the authors note, they have also been measured using single molecule FRET/nanosecond FCS, as for example in Soranno et al., 2012 (<https://doi.org/10.1073/pnas.1117368109>), where reconfiguration times of IDPs in the range of 50-100 ns have been measured, agreeing very precisely with the time ranges the authors report.
- The comparison of different relaxation rates and time scales is much improved in the revised manuscript, and I thank the authors for this immense effort. In order to fully take into consideration the different time scales sampled by an IDP, the authors should also briefly put their statements on an overall rotational correlation time into context, especially since IDPs are known to have dynamics on various rapid time scales.
- Figure 2B (comparison of R2 between MD and experiments) seems actually quite good, also for the first 10 residues. Deviations are between around residues 7 to 15 and in helix 2. Bigger deviations around the N-terminus appear to stem from different time scale motions. Can the authors explain this?
- I am not convinced of the interpretation of the PET-FCS data. Where do the authors get the approximate resolution of 40 ns from? The counting electronics used in this study usually have a much better timing precision. Just because the curve starts at around 40 ns, does not mean that the resolution is 40 ns. A 20 ns component that tails into the recorded curve should thus be picked up experimentally.
- It is not fully clear to me, why the simulations do not capture the correlation times (and R2 rates) around residues 23-27, which are much shorter than the length of their simulations. If the simulations can reach to time scales accessible by CPMG relaxation dispersion, should they not confidently pick up dynamics assessed by spin relaxation?
- Ntail is known to have non-random secondary structure. The authors should also take this into account in their discussion and cite the relevant literature, if they want to discuss deviations from simple homopolymer model predictions.
- Reference 66 does not use CPMG profiles (line 843). The authors should carefully revise the reference.

Reviewer #2

(Remarks to the Author)

I sincerely appreciate the authors' careful attention to the comments on the previous version of the manuscript. This version is much improved and suitably highlights the technical advances reported in this work and the novel insights gained into the dynamic landscapes of IDPs. I have no further suggestions and recommend this manuscript for publication without reservations.

Reviewer #3

(Remarks to the Author)

I thank and congratulate the authors for this second version of the work that makes it more accessible and clarifies the agreement between simulations and experiments.

Responses to Reviewers' Comments

Reviewer #1:

Reviewer Comment #1: The manuscript by Szöllősi et al. deals with dynamics of intrinsically disordered proteins (IDPs) across different time scales – an emerging topic of great interest and biological importance. The authors employ and combine advanced NMR spectroscopic techniques and molecular dynamics simulations to access time scales that have previously been difficult to access. For this, they make use of high power ^1H relaxation dispersion experiments that they have previously developed, combined with more classical NMR approaches, such as spin relaxation. In combination they thus experimentally tackle time scales in the picosecond to nanosecond time scale and in the low microsecond to millisecond time scale (and by extrapolation down to the hundreds of nanosecond time scale). Thereby, they decrease the gap of NMR accessible time scales, in which the IDP p53 shows dynamic features. Time scales faster than a few microseconds are accessible by long MD simulations and compared with the experimental data. Their combined model reveals hierarchical dynamics of chain motion, and transient folding of secondary and small tertiary structures that the authors think may be relevant for function. They assume that other IDPs will show similar functional time scales.

The study presented touches a highly important topic and presents sound data that suggest the presence of dynamics that has not previously been uncovered. However, the manuscript is hard to read, since it is composed of a very short main text with an extensive supplementary file and an additional extended data file that cross-reference each other. Likely due to this partially confusing structure, the conclusions and what exactly has been done is not always clear, and this needs to be revised along the following points:

Reply: *We thank the Reviewer for the very positive and thoughtful assessment of our work, and for recognizing both the significance of the topic and the strength of the experimental and computational approach, despite the difficulty of reading the original submission. Regarding the formatting of the originally submitted manuscript, and as indicated in our cover letter, we wholeheartedly agree that the presentation was close to undigestible, largely imposed by the rather rigid formatting requirements for our initial submission to Nature, which are much stricter than for Nature Communications. This required a very compressed main text and had forced us to split our Supplementary Material into three scattered parts which definitely made it hard and cumbersome to read. Being aware of this, we had submitted a cleaned up and de-fragmented version in parallel to the automated transfer of the manuscript from Nature to Nature Communications, which unfortunately has not made its way to the Reviewers for reasons outside our control. In addition to addressing the Reviewer's comments as detailed below, we have now re-arranged, and in part re-written, much of the manuscript, making best use of the flexibility and space offered by Nature Communications. In particular, we have now merged all results that are important for the logical flow of the narrative (and which were initially scattered across diverse supplements) back into the main text, which we think has improved the readability and flow considerably.*

We have also now combined all the original Supplementary Information / Extended Data parts into one, moved those sections and Figures into main text that were highlighted as important by Reviewers' comments, and streamlined the main text accordingly. Specifically, we have (1) moved Supplementary Section (B) ("Difference between relaxation from exchange versus relaxation due to molecular tumbling") to the main text, adding a new panel E to Fig. 2, (2) moved Supplementary Section (D) ("Uncertainties and Controls") to the main text and expanded the discussion of the comparisons to further experimental data, as suggested by Reviewers; (3) we have moved the Results and Discussion of the chemical shift comparison to the main text (now Fig. S11 remains in Supplementary Information, though); (4) we have moved parts of Supplementary Sections G ("Calculation of SDFs ..."), parts of Extended Data Section D ("Comparison of measured with calculated NMR observables η_{xy} , τ_c , R_1 , R_2 , and NOEs"), and parts of Extended Data Section (E) ("SDFs calculated from MD simulations; comparison with NMR measurements and a simple tumbling model") to main text, adding a summary Fig. 5 (the full data – Figures S6, S7, and S8 – remain in the Supplementary Information part).

With these re-arrangements of the text, we have also largely removed 'results/ discussion'-type descriptions that were in the Supplementary Information, and/or have moved parts of these to the main text where appropriate. Finally, it should now be much easier to find the methods sections related to the individual results, which we have all moved to the Methods Section after the main text.

Regarding readability, please also see our reply to the similar remark by Reviewer #2 below. We very much hope and trust that the Reviewers and potential readers of our manuscript will now find it much easier to read.

We also very much appreciate the careful reading by the Reviewer, and have implemented all suggestions, which clearly further enhanced the strength and readability of our manuscript.

Reviewer Comment #2: In their introduction, the authors compare dynamic time scales that they address by NMR spectroscopy with those that have previously been addressed by single molecule spectroscopy. They mention time scales in the nanosecond range (line 68) and that the time scales between 100 nanoseconds and 10 microseconds remain elusive. This is only partially true, approaches based on fluorescence correlation spectroscopy, which the authors also cite, are sensitive from the picosecond time scale down to the millisecond time scale without time scale interruptions. In fact, classical interconversion times that have been found in IDPs using FRET and FCS combinations are often in the range around 100 nanoseconds. This should be adapted in the text accordingly.

Reply: *We thank the Reviewer for pointing this out, and fully agree. What we meant to write, but didn't, is that while these timescales have been probed by different experimental techniques, most prominently FCS, structural detail of the underlying dynamics is lacking.*

In our revised version we now clearly state that combinations of spectroscopic experiments such as those described in Chowdhury et al. 2023 or Soranno et al. 2012 probed kinetics covering the broad range between 100 nanoseconds and 10 microseconds, and that our combined NMR/MD approach adds structural detail to these dynamics (2nd paragraph of the Introduction of the revised manuscript, and also line 325 and 1st paragraph of the Discussion).

Reviewer Comment #3: Related to this point, lines 103 to 111 require a more detailed discussion. IDP dynamics measured by single molecule fluorescence and NMR approaches rely on very different observables (e.g. distance fluctuations vs fluctuations in chemical shifts, which is much more local). This aspect needs to be discussed, because it affects the actual time scales expected and how they are interpreted.

Reply: This is certainly a valid point, and we have now refined and extended our discussion appropriately (lines 119-130). We note that even though the NMR chemical shift measurements are indeed very sensitive for local changes, they are still able to capture slower and larger cooperative motions, also because slower dynamics cause large NMR signals. We also note that the structural character of the underlying dynamics is accessed by our MD simulations, irrespective of the experimental observables used, and compared to the available experimental observables including FRET. In the revised text we have now noted this point.

Reviewer Comment #4: In Fig. S2, the authors show a comparison of ΔR_2 , which is the effective R_2 at the highest available CPMG frequency from which the R_2 calculated from TRACT experiments was subtracted. The figure shows that between the time scales accessible using high power CPMG experiments and the 'molecular tumbling time scale' further dynamics exist. The authors call this an 'otherwise blind spot'. However, also with their new experiments, they cannot access these time scales quantitatively. They should be more explicit about that. Furthermore, it is worth including the presence of these dynamics in some way into Figure 1G.

Reply: We agree that our use of the word 'otherwise' can be misunderstood in the way the Reviewer pointed out, and have now reformulated the text accordingly (lines 190-196). Specifically, we now state that while we are able to infer that these dynamics exist, their quantification was not possible. We have also adopted the suggestion of the Reviewer and have now moved the respective supplementary figure and discussion into the main text as Fig 2E, which improved the logical flow and makes this point more accessible for the readers.

Reviewer Comment #5: The authors claim that they can assess protein dynamics down to 80 ns (tier 1) using RD. However, this is only by extrapolation from very cold temperatures.

The authors should be a bit more clear on this. Furthermore, the extrapolation is based on experiments conducted at four different temperatures, from 262-265 K. These are very small differences compared to an exponential extrapolation to 298 K. The authors should discuss this aspect and the validity of their approximation.

Reply: *We thank the reviewer for highlighting the need to clarify the basis and limitations of our extrapolation. We agree that our extrapolation, based on the fitting to the Arrhenius equation, may indeed appear somewhat bold. The narrow temperature range accessible experimentally (262–265 K) was limited by two constraints, (i) the requirement that the exchange process remain within the detection window of NMR relaxation dispersion, and (ii) sample freezing at 261 K even under the supercooled conditions inside capillaries. To avoid over-interpretation of this extrapolation, we have used Bayesian inference to provide a rigorous uncertainty estimate, so that the predicted time scale uncertainty includes potential extrapolation errors. We have now added an improved explanation of this fact to the revised main text (lines 111-118 and lines 710-715). As an additional check, we have now also performed an extrapolation to room temperature using Eyring's formula, which, in contrast to the empirical Arrhenius equation, is based on transition state theory and describes the temperature-dependency of the attempt frequency (or prefactor). As we now also mention in the revised manuscript (lines 117-118), the resulting estimated room temperature rates are very similar [Arrhenius: $(83.6 \pm 18.4 \text{ ns})^{-1}$; Eyring: $(63.2 \pm 13.8 \text{ ns})^{-1}$], which does not change our conclusions. Note also that additional support for the tier 1 protein dynamics comes from the agreement between MD simulations at 298 K, which do show relaxation due to tier 1 dynamics, and levels at the same $R_{2,\text{eff}}$ as the 298 K NMR RD measurements (Fig. 2F-I and Figs. S4-S5 in the revised manuscript).*

Reviewer Comment #6: It remains unclear how the comparison of MD derived RD curves with the experimental data is performed. It seems that the experimental data plotted in Fig. 2E-H and Fig. S4 should be the same as those in Fig. 1C-E and Extended Data Fig. 1. However, this is apparently not the case. Both types of plots have at their y-axis R_{eff} (1/s), but the values of the experimental data points are dramatically different between Figs. 2E-H/S4 and Figs. 1C-E/Extended Data Fig. 1, and so are the shapes of the curves. Have the authors performed some operation with the data before comparing with the MD derived RD curves? Maybe this information got lost somewhere in between the three files (maintext, extended data and supplement). However, understanding this aspect is crucial for judging the implication of the comparison of time scales between NMR and MD. A clear explanation for how the data were treated is thus necessary to understand this aspect.

Reply: *Thank you for pointing out this confusion in our description. The seeming discrepancy arises from the different temperatures of the two measurement sets, which affects the amplitudes of the RD curves – so there is in fact no inconsistency. Our original captions of these Figures did not make this fact clear, which we have now corrected in the revised version. To help the Reviewer identify the figures in question and the data they*

depict, the table below summarises the relevant information both for the original and for the revised version of our manuscript (note that the numbering has changed).

Submitted version	Revised version	Data plotted
Fig. 2E-H	Fig 2F-I	Experimental data vs. MD data at 298K
Fig S4 (and S3)	Fig S5 (and S4)	Experimental data vs. MD data at 298K
Fig 1C-E	Fig 1C-E	Experimental data at 263K
Ext. Data Fig 1	Fig S1	Experimental data at 263K

Reviewer Comment #7: Related to the above point, the authors also mention some ‘high-power artifacts’ that increased the measured RD profiles at higher frequencies. Can the authors be more clear what artifacts they are talking about?

Reply: *Indeed, we observed that with increasing radiofrequency power the transverse relaxation rate increased but only in experiments at 298 K. The origin of this increase is unclear and most probably due to hardware properties since exchange contributions to the transverse relaxation rate should become smaller; also, if heating (which we don't observe) was the reason, it would also reduce the transverse relaxation rate. We could not really find out what the problem was, but since it only occurred at high power, we excluded those points from our analysis and focused on the lower radio frequency irradiation instead. We have now added an explanatory sentence to the revised main manuscript (lines 246-248).*

Reviewer Comment #8: Also related to the same point is the interpretation of Fig. 2B, which shows $R_{2,eff}$ at a CPMG frequency of 100 s⁻¹. The values agree with those plotted in Figs. 2E-H and Fig. S4, but not with those plotted in Figs. 2E-H/S4 and Figs. 1C-E/Extended Data Fig. 1. The authors should be very clear about what exactly they are visualizing.

Reply: *We have now added appropriate explanations to the caption and annotation to the respective Figures to avoid potential confusion regarding the seeming inconsistency discussed in reply to the above comment. We also explain now that these measurements were performed at a different temperature (lines 230-231).*

Reviewer Comment #9: In the Supplementary Information, the authors say that they compare MD derived orientational correlation times with the TRACT relaxation times and cite Fig. 2B. However, Fig. 2B apparently shows $R_{2,eff}$ at a CPMG frequency of 100 s⁻¹ (see above). Which is correct?

Reply: While the $R2,eff$ at room temperature does include a significant portion of relaxation due to tumbling, we agree that such comparison is indeed not the most direct way to compare measured TRACT relaxations with those calculated from orientations extracted from our MD trajectories. In line 303, we therefore now refer to revised Supplementary Information Fig. S6 panel R2, which shows a much more direct comparison.

Reviewer Comment #10: In the MD data analysis, the authors identify different Markov states of which they determine the interconversion. They start out with a certain number (15 for the wild type), which they then cluster further into 8. Can the authors be more explicit on how these states were defined given their proximity in the TICA space? What were the criteria for reducing 15 states into 8? And how does the initial choice of number of states affect the results?

Reply: We had indeed not justified these choices in the original main text. Briefly, we merged the macrostates from 15 into 7 heuristically and guided by the aim of grouping states that are adjacent in the 2D TICA projection, according to the number and strength of hydrogen bonds as highlighted in part C of the Figure. Regarding the initial choice of 15 states, we had actually tried 5,7,9,11,13, and 15 initial states and found that, although differing in detail, each attempt captures the main result, which is the step-wise folding of helix1. We eventually chose to start with the largest number 15, which provides a more fine-grained picture and also introduces less bias due to the particular placement of the state boundaries.

We have now added an appropriate explanation (lines 897-899) and a new Table 1 listing which states were merged to the Methods Section of the revised manuscript.

Reviewer Comment #11: The comparison between MD simulations and FRET distance distributions needs clarification: It is not clear, what equations the authors used to calculate distances from FRET efficiencies. Was the IDP nature (rapid sampling of distances) taken into account for example? How were the dyes (sometimes fluorescent proteins appear to have been used, which are quite big) considered in the comparison? The authors also talk about experimental and MD derived distance distributions, but only show a single distance for the experimental data. Can they clarify this? It would be helpful to transparently lay out the experimental conditions of every experiment that has been compared and explain the conditions under which the distances have been compared, respectively.

Reply: We agree with the Reviewer that our descriptions of both the experimental conditions (taken from the cited measurements by others) and how we calculated the FRET distributions were a bit scarce. In our revised version, we now provide all requested information in the revised Supplementary Information Section J(b) (1st paragraph). Specifically, to calculate the distances from the FRET efficiency, we used the reordered equation from Moses et al., as is now explained in the supplements. We did not include

explicit dye molecules within our simulations, and used the distribution of Ca-Ca distances instead. As a result of this approximation, one would expect to obtain slightly smaller average distances from the simulations in those cases where the linker orientations are anisotropic and tend to point away from the Ca atoms. We also assumed that the measured distances are ensemble averages, implying rapid sampling and that the errors reported in the cited literature are errors of the mean distance. In comparing these mean distances with the distance distribution that we obtained from our MD simulations, we aim to test the plausibility of the simulated ensemble also for those few cases for which the measured average distance deviates from the average distance calculated from our MD ensembles. In the plots (Fig. S13), we have now clarified the different nature of these distances and distributions by superimposing vertical lines and distribution histograms. As we now also explain in the revised Supplementary Information Section J(b) (1st paragraph), the dyes used in the cited experiments were Alexa fluorophores or naphthalene derivatives for the distances 10-56, 1-17, and 14-36 (panels A-C on Fig. S13), which are not particularly large and are considered to rather isotropically sample their configuration space, such that the average distance between these dyes should not deviate too much from the Ca-Ca distances. Also, their transition dipole orientation is expected to be rather isotropic, such that the $\kappa^2 = 2/3$ approximation should hold well. In contrast, the dyes applied by Moses et al. were fluorescent proteins (donor: mTurquoise2 and acceptor: mNeonGreen) which, as the Reviewer pointed out, are rather large compared to the tested peptide length (1-61, panel D on Fig. S13), and are also expected to interact more strongly with the IDP.

We have therefore now adapted the discussion in the revised Supplementary Information Section J(b) (2nd paragraph) to better explain these limitations both on the experimental as well as on the computational side, and have added all experimental and computational details required to evaluate our assumptions and to replicate our approach.

Reviewer Comment #12: The authors point out the hierarchical time scales they find in an IDP and a 'folding funnel' similar to folded proteins (line 235). In fact, and this is what also their comparison of long lived structural elements with the PDB shows, the authors are identifying small (transiently) structured elements within the IDP – and these are subject to hierarchical time scale motions. This seems not to be the case for fully disordered regions with no secondary structure. This also goes along with the notion that a continuum exists between disordered and folded proteins. The authors should discuss this aspect.

Reply: *We fully agree with the Reviewer's observation, which indeed points to the interesting fact that the hierarchical timescales we observed are distributed quite heterogeneously along the sequence of the IDP. For example, for some segments of the protein (such as the terminal parts), no timescales are seen other than the fast reorganisation dynamics. Interestingly, there are parts of the IDP for which transient tertiary structures are seen outside the helix 1 and helix 2 regions, indicating that secondary structures are not always required for transient tertiary structure formations and correspondingly slow tier 0 dynamics. We thank the Reviewer for pointing out that this*

finding is worth discussing, which we now do in the main text (lines 413-421 and 567-571). We also have combined the new plot shown below with Fig. 3E of the original manuscript into a new Figure 4 of the revised main text, which now provides quantitative data to the discussion that support and illustrate the two above main points.

Reviewer Comment #13: The above point also relates to the discussion section, where the authors describe that their work ‘challenges the prevailing view that the lack of stable folded structures of IDPs implies an unstructured and shallow free energy landscape that only gives rise to [...] tier 2 [...] dynamics’. It is now accepted that a continuum of folded and disordered structures exist and that many IDPs possess transiently folded elements. This view does not take away any of the novelty of the presented manuscript, since slower timescale motion have indeed not often been observed in unbound IDPs by NMR spectroscopy and certainly not on the extended time scale presented here. However, this aspect has to be discussed in a balanced manner. Furthermore, citation of Nettels et al. at this position seems somewhat misplaced. In that article, dynamics on the order of 40-70 ns have been found, which might much rather correspond to tier 2 (or in between tier 1 and tier 2). In addition, the article deals with a guanidinium hydrochloride unfolded protein, which will certainly sample dynamics associated with transient structures to a much lesser extent than an actual IDP. The authors should consider this in their revision.

Reply: We agree with the Reviewer and appreciate his/her precise summary of the novelty of our findings. We have now removed a somewhat biased claim from the abstract (lines 47-48) and have rewritten the respective parts of our discussions (lines 572-584), which indeed made the discussion more balanced and now better reflects the state of the art. We have also adapted the citations of Nettels et al. as pointed out, and we now also mention further previous work that helped to present a fair perspective (lines 527-532).

Reviewer Comment #14: The idea that Measles Virus N_{TAIL} possesses similar hierarchical dynamics as p53 is appealing. However, experimental data would be needed to support

this claim – especially since the authors point out that the choice of the force field is strongly dependent on the protein under study.

Reply: We appreciate the positive remark by the Reviewer, and have now added to our evidence from atomistic N_{TAIL} simulations a reference [Otteson et al. (2025), 10.1038/s42004-025-01682-0] to experimental work (lines 552-554 of the revised main text, and Section (O) in the revised Supplementary Information, Fig. S20) that supports our findings. Specifically, Fig. 1 Panel D of this paper [Otteson et al. (2025)] shows measured dynamics on timescales between 2.2 and 8.2 μ s and a comparison to predictions from a self-avoiding wormlike chain model. The unexpectedly high τ_{cw} that clearly deviates from the model (and also regarding secondary structure formation) but agrees with our atomistic simulations provides experimental evidence that also our simulations of this different protein capture its dynamics sufficiently accurately to support the observed broad range of stretching parameters (revised Supplementary Information Section O, Fig S20B) as the main evidence for hierarchical dynamics. Consistently, and independent of any atomistic simulations and force field challenges, the deviation of the measured high τ_{cw} from the wormlike chain model prediction, as well as the observed slow time scales, provide evidence for complex N_{TAIL} dynamics.

Minor points:

Reviewer: The P27A mutant is only mentioned somewhat on the side. A dedicated paragraph to introduce this mutant and what is expected from it would benefit the readability of the manuscript.

Reply: We agree and have now made better use of the extended space that Nature Comm. allows for the main text. Among other explanations, we now specifically better explain the original rationale of the P27A mutant (lines 141-147 in the main text of the revised manuscript). Note that P27A does not occur naturally and was designed only for the experimental purposes now described in the revised main text (also lines 141-147).

Reviewer: The spectral density function is sometimes referred to as ‘SDF’ and sometimes as $J(\nu)$. A uniform expression should be adapted throughout the manuscript.

Reply: We have adopted the Reviewer’s suggestion and now generally refer to the spectral density function as “SDF” if it is the subject of a sentence and, occasionally, as “ $J(\nu)$ ” in mathematical expressions or whenever its argument (frequency) is important. As a result, we used a uniform expression, as suggested, almost everywhere, with few exceptions where readability would be impacted (e.g. lines 445 or 456). We are of course happy to adapt to Journal style.

Reviewer: The authors state that ‘contributing to the NMR RD profiles are also faster, ns and ps motions...’. This sentence merits another phrase to describe how these change the relaxation dispersion curves.

Reply: *Thanks for flagging this. We now added a phrase explaining that these faster motions add an offset to the relaxation dispersion curves (line 425) and also now mention in lines 272-273 that the RD curves calculated from the MD simulations do not contain an $R_{2,0}$ offset arising from tumbling of the molecule.*

Reviewer: In the context of the analysis of $J(\nu)$, the authors state that there is again evidence for ‘multi-state dynamics involving a wide spectrum of timescales also within the fast tier 2 regime’ (lines 314-315). This agrees with dynamics that have been observed for other IDPs and should be supported by a small discussion along these lines.

Reply: *As suggested, we have now added a brief discussion pointing to previous measurements (lines 488-491).*

Reviewer: The statement that ‘such transient structures are not merely biochemical curiosities’ is maybe a bit too general, given that the authors have studied one protein. Maybe they can tune this expression down a bit.

Reply: *We have now tuned down the expression, as suggested (lines 578-581, revised manuscript main text). It now reads: “Knowledge of such transient structures may help shift their conformational ensembles and modulate their activity, such that these structures may become potential drug targets.”*

Reviewer: The authors mention ^1H off-resonance $R_{1\rho}$ data in the methods section. It is unclear, where these data can be found in the manuscript.

Reply: *Many thanks for pointing this out. In the revised manuscript, we have now added an additional Supplementary Information Section “(B) $R_{1\rho}$ raw NMR data” and a new Fig. S2, which shows the full raw data. We have now also improved the description in the Methods section (lines 654-658, revised manuscript main text) to better explain our approach. Briefly, the backbone amide proton ($^1\text{H}_\text{N}$) $R_{1\rho}$ measurements between 262 K and 265 K was the basis for the Arrhenius extrapolation now presented in Fig. 1F in the revised main text.*

Reviewer: The authors should add a proper figure legend to Fig. S4.

Reply: To save space, in the original manuscript the legend of Fig. S4 (now Fig. S5) referred to the legend of Fig. S3 (now Fig. S4, revised manuscript, Supplementary Information Section D “Comparison of NMR and MD RD profiles”), which presents the data in exactly the same manner, just for the mutant instead of the WT. We have now replaced the reference by an explicit description of the Figure, as requested.

Reviewer: Figure S7: Is there any particular reason, why the authors compare wild type experimental data with P27A simulated data?

Reply: We think here the Reviewer may have been misled by our wording in the caption to Fig. S7 (now Fig. S9 in the revised manuscript, Supplementary Information Section G “Effects of additional residues at the N-terminus”) and the explanation right above this Figure in the original manuscript, which indeed first mentioned the comparison of WT NMR and P27A calculated from the MD simulations. But in fact, the primary comparison, aiming at assessing the effect of the N-terminal extension required for cloning, is between P27A (orange) and P27A-extended (magenta), both calculated from the MD simulations. We chose to use the P27A mutant for this comparison, because it required more (four, GSHM-) extra residues for cloning than the WT (only two residues, GS-), so that is the worst case and therefore should provide an upper bound for the expected effect. Indeed, as expected, clear differences are seen for N-terminal residues up to Val10. The second comparison addresses the question if for N-terminal extensions the MD simulations can reproduce the measured SDF profile better than for those without extension. Here, SDFs were only available for the WT-extended. A similar effect on the N-terminal dynamics is seen as for the simulations that include the extension, and a clear difference compared to those without, further corroborating our conclusion. A third comparison is between differently protonated constructs (magenta and purple) to assess the effect of the uncertain protonation states (result: very small). Of course we could have performed additional simulations of the extended WT, but we felt those would not have provided additional evidence for our main conclusion that the N-terminal extensions are a major contributing factor to the observed N-terminal differences between the measurements and simulations discussed in the main text.

We have now re-written the caption to Fig. S9 and parts of the explanatory text in the respective revised Supplementary Information part right above the Figure (mainly 2nd paragraph) to avoid this confusion.

Reviewer: Figure S8: Can the authors comment, why the R2 is not reproduced so well and how this may affect the interpretation of their data? How this related to the other data comparisons undertaken? The green lines in the figure should also be labeled with the force field used under these conditions.

Reply: The primary purpose of this Figure (now Fig. S18 Supplementary Information Section M “Comparison of different force fields” in the revised manuscript) is to assess the

accuracy of different force fields, corroborating that the one we chose for the large-scale production runs (Amber), is the most accurate. Indeed, R_2 is reproduced less accurately than the other NMR observables (although we would maintain that it still reproduces the main features of the respective profile quite well). Our best guess is that, in contrast to the other observables, R_2 is determined essentially by all parts of the SDF and therefore more sensitive to any inaccuracy anywhere on the frequency spectrum. As can be seen in Fig. S7 Supplementary Information Section F “Comparison of SDFs calculated from MD simulations with NMR measurements” of our revised manuscript, the SDFs calculated from our MD simulations are more accurate at $\nu_1 = \omega_N/2\pi = 96.3$ MHz than at the other frequencies, which may explain the discrepancy. We have now improved the label to the green line in the Figure as suggested (thank you!).

Reviewer: Figure S9: It is interesting to see that the MD derived RD profiles all level off at an $R_{2,eff}$ of 0. This is not the case in Fig. 2E-H and Fig. S3. Can the authors comment on this?

Reply: This is a valid observation. The reason is that while the RD profiles on Fig. 2E-H (now Fig. 2F-I in the revised manuscript, main text) and Fig. S3 in the original manuscript are the composite $R_{2,eff}$ (which is the sum of exchange related relaxation and relaxation offset due to fast reorientation motion), Fig. S19 revised Supplementary Information Section N “Convergence analysis of the MD simulations” focuses on the convergence which is dominated by the slow dynamics (exchange related relaxation). Therefore, the profiles in revised Fig. S19 show only the exchange related relaxation without the fast reorientation related relaxation. In the revised manuscript, we have adjusted the y-labels, captions, and text (lines 269-278, main text) to clarify this issue.

Reviewer: Extended Data Figure 2 shows chemical shift comparisons between MD derived and experimental chemical shifts. How do these plots look like when secondary chemical shifts are compared?

Reply: We agree this is helpful information and have therefore now included the plot below as Fig. S11E-H within the revised Supplemental Information Section I “Comparison of measured chemical shifts with those calculated from the MD structural ensemble”. Of course, the result depends on the used reference value of the random coil. For the plot below, we have used reference values used by SPARTA+, which depend on residue and atom type; other methods additionally consider neighboring residues (Kjaergaard et al. (2011) [10.1007/s10858-011-9472-x](https://doi.org/10.1007/s10858-011-9472-x)). Not unexpectedly we find that the mean absolute error remains unchanged, whereas the Pearson correlation coefficient decreases somewhat.

Reviewer: Extended Data Figure 3: What are the horizontal lines in the experimental radii?

Reply: *The horizontal lines indicate the experimental error reported in the respective publications that are mentioned in the insets. We have now adjusted the figure caption (now Fig. S12 in the revised Supplementary Information Section J(a) “Ensemble averaged radius of gyration and hydrodynamic radius”) appropriately. Thank you for flagging this!*

Reviewer: Extended Data Figure 5: Considering the comparison of MD with PET-FCS measurements, it would be interesting to see until which time point exactly experimental data have been obtained, and compare this time point with when the black and cyan line start deviating from each other.

Reply: *Unfortunately, the precise time window of the experiment was not explicitly mentioned in the respective publication [10.1021/ja2078619]; we therefore estimated the*

fastest measured timescale from the respective plots to be ~40 ns, which is also the lower end of our x-axis in (now) Fig. S14 of the revised Supplementary Information Section J(c) "Photoinduced Electron Transfer Fluorescence Correlation Spectroscopy (PET-FCS)". As can be seen, the reproduced experimental curves without and with the hypothetical 20 ns process (black and cyan lines) start to deviate already around 100 ns. This behavior is expected for the hypothetical 20 ns process used in our analysis.

Reviewer: Extended Data Figure 8: Can the authors explain the difference between the mutant NMR and MD data?

Reply: Note that in our revision we have moved parts of this figure to the main text as Figure 5C and the whole Figure as Fig. S6 to the Supplementary Information Section E "Comparison of measured with calculated NMR observables η_{xy} , τ_c , $R1$, $R2$, and NOEs") The main differences are indeed seen for the mutant, and particularly for residues 23-27, most likely due to the pronounced tier 0 dynamics of P27A, which may not be sufficiently (and with sufficiently many transitions) sampled by our MD simulations to achieve fully converged results; see also our cautionary note to this effect (lines 269-278 and lines 395-398 of the revised manuscript main text). Also, while we used for the calculation of $J(\nu)$ (the basis to derive the other values) the longest possible autocorrelation window size (10 μ s) the experimental timeframe is much wider, such that much slower conformational excursions may be missed. We have now added these explanations to the revised manuscript main text (lines 481-491).

Reviewer: The authors indicate supplementary movies. Where can they be found?

Reply: The movies have been uploaded to the github page of the project and are readily available: https://github.com/dszollosi/p53_TAD_dynamics. This is also now mentioned in the Data and materials availability section of the main text.

Reviewer #2:

Reviewer: This manuscript from the Griesinger and Grubmuller labs describes a recent collaborative effort to decode the dynamics of transient structure formation in the well-studied p53 N-terminal transactivation domain (TAD). Using high-powered relaxation dispersion methods and extensive molecular dynamics simulations, the authors obtain new insights into the identity of transiently formed secondary and tertiary structures in the p53 TAD and the timescales by which these structural elements interconvert. The experiments

and analysis described in this work are rigorous and thorough and there is little to criticize about the technical aspects of this manuscript.

My enthusiasm for the manuscript overall is unfortunately dampened by the presentation of the data and the brevity of the text. As currently written and organized, the authors have failed to present the data in a manner that allows them to highlight the exciting potential of the state-of-the-art methods they have used here. In general, I struggled to understand why so much information was relegated to the Extended Data and Supplementary Information. Much of the text included in the Supplementary Information is absolutely essential for the reader to be able to interpret any of the results presented in the main text. I hope that the authors will make significant efforts to rewrite this manuscript in a way that thoroughly explains why certain experimental and computational methods were used, how the results from these experiments can be analyzed, and what sort of interpretation is possible. In its current iteration, the impact of the work is totally lost.

Reply: *We thank the Reviewer for the very positive and encouraging assessment of the scientific content of our manuscript, and for the frank words on the presentation. Regarding the presentation, and also in line with our reply to the similar comments by Reviewer #1, we can only wholeheartedly agree with the observation that the severe length limitations set by our original submission to Nature (which we can understand), combined with the similarly severe limitations and rules regarding Supplementary / Extended / Methods material split over three text bodies (which we cannot really understand), made quite a difficult reading. Unfortunately, for reasons outside our control, the streamlined and re-arranged version we had originally provided during the automated transfer from Nature to Nature Comm. was not provided to the Reviewers, which would probably have made the job of the Reviewers much easier. As you will see, we have indeed made significant efforts to improve the readability of our manuscript, and our Revision has now been largely re-written and re-arranged to avoid such unfortunate scatter, additionally now taking all Reviewers' comments into account. For a more detailed description of the major changes and re-arrangements we made, please see our reply to Comment #1 by Reviewer #1. We very much hope and trust that the Reviewers and potential readers of our manuscript will now find it much easier to read.*

Reviewer Comment #1: When the authors present the results of the relaxation dispersion experiments, they do not explicitly state why some data are not shown. This is particularly notable in Fig 1E where there is not a green curve shown despite the legend stating that there should be, but this is pervasive throughout. Presumably data are not shown for residues that either did not show dispersion or for residues that were not assigned and/or did not have detectable resonances in the spectra. This should be explicitly stated and explained.

Reply: *We agree that the caption of Fig. 1E in the original manuscript was misleading, which we have now clarified. Indeed, as the Reviewer assumes, data are not shown for*

those residues for which no dispersion was seen in the NMR data, which is now described in the revised legend to panel E of Fig. 1.

Reviewer Comment #2: Fig 1G – the WT data points for the Tier 2 dynamics are hard to decipher as they are largely hidden by the data for the P27A mutant.

Reply: *Thanks for spotting this one – we have now replaced the green WT ‘x’ by ‘+’ in the revised version of Fig. 1G to render them better visible. Also, like the RD data points, we have now moved the green ‘+’s to the foreground, further enhancing their visibility.*

Reviewer Comment #3: The color scheme in Fig 2 is problematic. It is difficult to distinguish between the light and dark green in these figures. Please consider using colors with higher contrast.

Reply: *We have now shifted the light green in the revised Fig. 2 towards cyan to increase contrast, such that it can be distinguished better from the dark green. We have, however, refrained from too much of a change, as the similarity of color tones is intended to communicate the grouping into WT and P27A mutant. We think with this change we have reached a good trade-off between graphics language and proper distinguishability.*

Reviewer Comment #4: The discussion of the stretched CPMG model in the main text is inadequate. What is the reader supposed to glean from the stretching factor and why this model would be appropriate to describe the data?

Reply: *We have now added an appropriate discussion to the main text of the revised manuscript (lines 367-375) and a reference to the explanation in the revised Supplementary Information Section H “Fitting CPMG model to RD profiles calculated from MD simulations” and Fig. S10. Briefly, as also discussed regarding Fig. 3D, stretched CPMG models, very much like the original stretched exponentials suggested by H. Frauenfelder, and in line with Ref. [Palmer [10.1021/cr030413t], Edholm and Bloomberg [10.1016/S0301-0104(99)00349-3], Blackledge [10.1021/acs.jpcllett.6b00885]], model a superposition of log-uniformly distributed relaxation rates (see also the illustrative Fig. S10E and its caption in the revised Supplementary Information Section H). Such a model is therefore a generalisation of the original CPMG model, which assumes a single relaxation rate, and, as our fits show, turns out to be a more adequate description of IDP dynamics. We are not aware of a particularly revealing interpretation of the specific value of the stretching factor beyond the general notion that the more the stretch factor deviates from 1, the broader the distribution of relaxation time scales in the frequency domain, except that the relaxation rates are the eigenvalues of the master equation that defines the Markov model [N. G. van Kampen, “Stochastic Processes in Physics and Chemistry” (1st ed. 1961)].*

Reviewer #3:

Szöllősi, Pratihar et al. report a combined NMR and MD study of the p53-TAD, an intrinsically disordered region (IDR). In particular, they focus on characterizing the different timescales at play in the system as observed by relaxation dispersion NMR and modeled by MD. They find that this IDR is characterized by a complex hierarchy of timescales ranging from sub-nanoseconds to tens of microseconds, associated with structural rearrangements spanning from side-chain motions to tertiary structural fluctuations. They further suggest that this hierarchy of timescales resembles that of natively folded proteins and may underlie the ability of this region to interact with multiple partners. The paper provides an interesting and detailed picture of the structural dynamics of p53-TAD; however, some of the claims appear too strong given the current data.

Reviewer Comment #1 Most of the timescales are measured under supercooled conditions to access fast dynamics and then extrapolated. It is not entirely clear whether the slower timescales were also probed at temperatures closer to physiological conditions.

Reply: We agree that our manuscript reports NMR data for supercooled conditions, but we also report NMR RD measurements at 298 K (e.g., Fig. 2F-I and Supplementary Information Figs. S4-S5 of the revised manuscript), which probe the slower timescales at 298 K. We think the combination of both, as well as the agreement obtained with room temperature MD simulations, adds to the strength of our manuscript. This comment, as well as comment #5 by Reviewer #1 (see above) made us aware that the temperatures at which the NMR measurements were carried out were not always clearly stated in the original manuscript, which we have now corrected (see also our reply to comment #5 by Reviewer #1). We also made sure now that the figure captions consistently show this information.

Note also that additional support for the tier 1 protein dynamics comes from the agreement between MD simulations at 298 K, which do show relaxation due to tier 1 dynamics, and levels at similar $R_{2,eff}$ values as the 298 K NMR RD measurements (Fig. 2F-I and Supplementary Information Figs. S4-S5 of the revised manuscript).

Reviewer Comment #2 The slow (“tier-0”) timescale is observed only for the P27A mutant. Its interpretation and extrapolation to the wild type through MD simulations are used to discuss the system as a whole. This is certainly an interesting observation, but claiming a hierarchy spanning seven orders of magnitude as a general property seems overstated given the current evidence (low temperature and mutation dependence).

Reply: Whereas it is true that the slow (“tier-0”) timescale is seen only for the P27A mutant by NMR, it is seen for both the mutant and the WT in our atomistic simulations. Given the good agreement between NMR and our simulations for many complementary observables, it would thus be very surprising if the (“tier-0”) dynamics were actually absent in the WT. In fact, the lower population of transient tertiary structures seen in the simulations of the WT

compared to those the mutant even predicts that the (“tier-0”) dynamics should be below the signal-to-noise level of the NMR measurements, as we have mentioned in the Methods part (see Section “Identification of metastable tertiary structure elements” in the Methods Section of the original manuscript and also in the revised manuscript main text). We have now added a new Fig. 4B in the revised manuscript to quantify these population differences. The Reviewer also rightly points out that this work mainly focuses at p53; however, to explore the possibility that comparably rich and complex dynamics may also be present in other IDPs, we also reported measurements and our atomistic simulation results for the Measles virus NTAIL IDP in the revised Supplementary Information Section O “Stretching parameters obtained from RD profiles for the Measles NTAIL peptide” , which indeed supported this notion [Otteson et al. (2025) <https://doi.org/10.1038/s42004-025-01682-0> and Fig 1D therein]. We think that this finding does provide some evidence that the observed hierarchy of time scales may indeed be a feature of other IDPs, albeit (agreed) not necessarily a general feature. We now mention this finding in our conclusion section (lines 551-553 in the revised manuscript main text). We agree with the Reviewer that our statement “[...] which suggests that these unexpectedly complex dynamics may be a general feature of many other IDPs” in the original manuscript may have been too strong, which we therefore have now adapted to avoid any claim of generality and thus to more adequately reflect the available evidence (lines 553-554 revised manuscript main text); it now reads: “[...] suggest that these unexpectedly complex dynamics may be a feature of other IDPs”.

Reviewer Comment #3 Following up on the previous point, the simulations do not reproduce the experimentally observed $\sim 215 \mu\text{s}$ slow exchange process in the mutant, instead identifying tertiary elements persisting only up to $\sim 5 \mu\text{s}$. It is therefore not accurate to state that the simulations “accurately reproduce” the experimental hierarchy. A stronger validation would result from hypothesizing and testing a mutation derived from the conformations identified by the simulations.

Reply: Note that the $\sim 215 \mu\text{s}$ dynamics mentioned by the Reviewer was probed at supercooled conditions, whereas the $\sim 5 \mu\text{s}$ persistence durations of tertiary structure elements were observed in room temperature MD simulations. Rescaled via Arrhenius’ law as described in the revised manuscript (lines 111-113 and Fig. 1F main text), the $\sim 215 \mu\text{s}$ dynamics would translate to $\sim 4.3 \mu\text{s}$ dynamics, which in our view is a good agreement indeed, given the uncertainties now also described in the revised main text (lines 114-118) and lines 668-670). As a note of caution, as we also discuss in the revised main text (lines 263-303 and 395-398), there are several uncertainties involved, the two largest of which may ‘pull’ the result in opposite directions. First, the $\sim 5 \mu\text{s}$ persistence durations cannot be directly translated into a relaxation rate, and, therefore, the actual formation rates may be slower. Second, the μs persistence durations that we have actually observed in the MD simulations are a lower bound, as it is very likely that rare, longer-lived tertiary structures have been missed in our still limited total simulation length. Combined, and given that no fit parameters were used, we would think that the reported comparison can be described as

remarkably good, and respectfully disagree with the Reviewer's claim that our simulations would "not reproduce experimentally observed ~215 μ s slow exchange process". Given the above remaining uncertainties, we have now changed our wording from "accurately reproduce" to "[...] sufficiently accurately to allow identification of the structural motions that give rise to the observed tier 1 dynamics." (lines 315-317).

Regarding the Reviewer's suggestion to provide further validation of our simulations by hypothesizing and testing a mutation derived from the conformations identified by the simulations, we would be happy to give it a try; however, the simulations for the WT and P27A took us three years in total, which renders such validation attempt, unfortunately, impractical. We would like to note, however, that our simulations actually predicted that the slower dynamics originally revealed by NMR for the P27A mutant is NOT a slowed down version of the tier 1 dynamics seen for the WT, as was originally assumed, but rather a new type of dynamics with a different structural substrate. This prediction actually triggered the subsequent 1.2 GHz NMR measurements which then proved that our prediction was indeed correct. We think that this successful prediction does provide independent additional validation of our simulations, and in a similar spirit as suggested by the Reviewer.

Reviewer Comment #4 A χ^2 value quantifying the agreement between simulations and SAXS data is not provided.

Reply: As suggested, we now provide a χ^2 value in our revised manuscript (Supplementary Information Section J(a) "Ensemble averaged radius of gyration and hydrodynamic radius", last paragraph), which turned out to be 0.978 and 0.946 for the WT and P27A, respectively using the 3.2 mg/mL experimental result as reference.

Reviewer Comment #5 The PRE comparison is performed using C α -C α distances, which is a rather low-resolution approach given the availability of more accurate methods such as DEER-PREdict (10.1371/journal.pcbi.1008551).

Reply: We thank the Reviewer for this suggestion and now also provide a comparison to PRE curves calculated using DEER-PREdict in the revised manuscript. As can be seen from Fig. S15 in the revised Supplementary Information Section J(d) "Paramagnetic Relaxation Enhancement (PRE)", the shape of the PRE profiles are captured well by both methods.

Reviewer Comment #6 The Markov state model lacks analysis of implied timescales and other tests that would demonstrate the robustness of the model.

Reply: As requested, we have now added Fig. S16A in the revised Supplementary Information in Section K "Markov state model analysis: Convergence and helix 2 folding dynamics", showing near-convergence of the implied time scales. As described in reply to

Comment #10 by Reviewer #1, we have also constructed Markov models with 5, 7, 9, 11, 13, and 15 initial states and found that, although differing in detail, each attempt captures the main result, which is the step-wise folding and unfolding of helix1. It is important to note, however, that we do NOT use our Markov model to infer implied (long) time scales that are not covered by the MD simulations (as, e.g., was necessary in the A β 42 MD paper [Löhr et al. (2021) 10.1038/s43588-020-00003-w] referred to by the Reviewer below). Rather, we use it to extract rates from the simulations which are faster than the trajectory length and, therefore, are accurately sampled. Accordingly, here we do not need to claim that the Markov model fully and quantitatively captures the dynamics of the helix; rather, we only use it as an analysis tool to extract the main structural intermediates along the reversible folding pathway, establishing that folding proceeds along successive formation of α -helical hydrogen bonds. We have added an appropriate note to the revised main text (lines 345-348) to clarify this issue.

Reviewer Comment #7 SAXS, FRET, and PRE data should be made available. While the trajectories are extensive, suitably pruned versions should also be shared for reproducibility.

Reply: We appreciate this suggestion and have now deposited the subsampled (every 10 ns) trajectories to the Edmond repository <https://doi.org/10.17617/3.JWVPWJ>. Note that the SAXS, FRET, and PRE data are not ours and are published as cited in our manuscript; to facilitate proper reproducibility, we have therefore now added those data that we have extracted from these publications and used for our analysis as part of the Source Data accompanying the manuscript.

Reviewer Comment #8 In terms of broader context, slow conformational dynamics on the tens-of-microseconds timescale have previously been suggested using MD simulations for A β 42 (10.1038/s43588-020-00003-w), and a similar picture emerged in terms of free-energy barriers for A β 40 (10.1038/srep15449).

In conclusion, this is an interesting and valuable study providing important insights into the conformational dynamics of an intrinsically disordered protein. However, some claims should be moderated to avoid unsubstantiated overstatements, without detracting from the significance of the work. A validated prediction related to the tier-0 state would dramatically strengthen the manuscript.

Reply: We thank the Reviewer for this positive assessment and for underscoring the importance of our findings. As suggested and detailed in our reply to comment #2 above, and also in line with comments by Reviewer #1, we have now adapted our conclusions to more accurately reflect previous work and the level of evidence that we provide. Regarding validated predictions about the tier-0 states, we refer to our reply above in the context of the measured $\sim 215 \mu\text{s}$ vs. $\sim 5 \mu\text{s}$ dynamics, where we explained that the appearance of a new type of structural dynamics different from helix formation dynamics, was actually a

prediction that was confirmed by subsequent NMR measurements. Along similar lines, the transient tertiary structures revealed by our simulations predict NOEs that are below the noise level set by current NMR technology, but in principle might be validated as soon as higher signal/noise levels can be achieved.

We also thank the Reviewer for pointing out literature which we now cite properly (lines 75-86 and 574-575, where we now also refer to the notion of an inverted free energy landscape). It is worth mentioning, though, that trajectories reported in the A β 42 paper are between 10 ns and 90 ns, such that the microseconds timescales reported there had to be derived indirectly via Markov modelling. In contrast, as also explained in reply to comment #6, the tier 0 tertiary structure formations reported in our manuscript are directly seen in our 30 μ s and 60 μ s trajectories and therefore do not need to be inferred or extrapolated.

REPLIES TO REVIEWERS' COMMENTS

Reviewer #1 (Remarks to the Author)

The authors have made a tremendous effort to simplify the flow of the manuscript and therefore significantly strengthened it. The authors stress that different hierarchical time scales exist also in IDPs.

Reply: We thank the Reviewer for this very positive comment.

Reviewer Comment #1: While this is interesting, it is important, particularly in the last paragraph of the discussion section, to make sure to distinguish dynamics sampled by IDPs from those of folded proteins.

Reply: We have now worked over the manuscript and particularly the last paragraphs of the Discussion Section to avoid any confusion with the dynamics sampled by folded proteins. We also have re-checked the references to make sure they refer to IDP where appropriate.

Reviewer Comment #2: The figures in the supplementary file are not cited in order, maybe this could still be adapted. In addition to this, only few comments remain to addressed.

Reply: We have now reordered the supplementary figures to match the order in the main text, thank you for pointing this out!

Reviewer Comment #3: Comparison of the different time scales in the results sections (lines 119-130) still appears somewhat incomplete. Both single molecule FRET as well as NMR are sensitive to fast nanosecond motions, which are much faster than the measured supra tauc motions. While similar motions have been measured using PET-FCS, as the authors note, they have also been measured using single molecule FRET/nanosecond FCS, as for example in Soranno et al., 2012 (<https://doi.org/10.1073/pnas.1117368109>), where reconfiguration times of IDPs in the range of 50-100 ns have been measured, agreeing very precisely with the time ranges the authors report.

Reply: We thank the Reviewer for pointing us to this finding, which we now properly mention and discuss in the respective paragraph (lines 117-135 in the colored revised manuscript).

Reviewer Comment #4: The comparison of different relaxation rates and time scales is much improved in the revised manuscript, and I thank the authors for this immense effort. In order to fully take into consideration the different time scales sampled by an IDP, the authors should also briefly put their statements on an overall rotational correlation time into context, especially since IDPs are known to have dynamics on various rapid time scales.

Reply: We agree and have improved the respective discussion (lines 187-196 in the revised manuscript) accordingly. We have also added a respective note saying that for IDPs no well-defined overall tumbling time exists.

Reviewer Comment #5: Figure 2B (comparison of R2 between MD and experiments) seems actually quite good, also for the first 10 residues. Deviations are between around residues 7 to 15 and in helix 2. Bigger deviations around the N-terminus appear to stem from different time scale motions. Can the authors explain this?

Reply: Unfortunately we do not have a definite explanation, and also agree with the Reviewer that the overall agreement is quite good. The segment between residues 7 to 15 is particularly rich in proline residues (Pro8, Pro12, Pro13) and due to the practically fixed isomeric state we might miss a certain exchange modes that are present in the experiment. This type of uncertainty is mentioned in the main text at lines 267-271, where also control simulations are reported. Regarding helix 2, there is only a single proline residue (Pro47) that might similarly cause deviations, but it might also be a force field issue, both of which we mention at lines 216-226 and 230-235 of the revised manuscript.

Reviewer Comment #6: I am not convinced of the interpretation of the PET-FCS data. Where do the authors get the approximate resolution of 40 ns from? The counting electronics used in this study usually have a much better timing precision. Just because the curve starts at around 40 ns, does not mean that the resolution is 40 ns. A 20 ns component that tails into the recorded curve should thus be picked up experimentally.

Reply: We thank the reviewer for pointing this out. We had requested the original data from the corresponding author of the respective publication (Lum et al.), but did not get a response, so we resorted to the reported values and the figures, which to us, being non-PET-FCS experts, suggested 40 ns resolution. We have now corrected the supplementary text accordingly.

Reviewer Comment #7: It is not fully clear to me, why the simulations do not capture the correlation times (and R2 rates) around residues 23-27, which are much shorter than the length of their simulations. If the simulations can reach to time scales accessible by CPMG relaxation dispersion, should they not confidently pick up dynamics assessed by spin relaxation?

Reply: Indeed, the correlation times (<2 ns) are well within the accessible timescales of our MD simulations and, as the Reviewer assumes, should in principle be fully captured. However, this is only true if all protein conformations were actually fully sampled in our MD ensemble, so here the timescale problem comes back in 'through the backdoor'. For illustration, imagine that our structure ensemble missed a few extended conformations. These would slow down the ensemble averaged correlation times and therefore, our correlation times calculated from the MD simulations would underestimate the measured ones. We think this is the likely cause of the deviation pointed out by the Reviewer, particularly for the P27A mutant for which transient tertiary structures are more frequent. In

short: the timescales are covered, but not the ensemble. Because we do not have sufficient evidence for this explanation, however, we would prefer not to include it into our discussion.

Reviewer Comment #8: Ntail is known to have non-random secondary structure. The authors should also take this into account in their discussion and cite the relevant literature, if they want to discuss deviations from simple homopolymer model predictions.

Reply: *We thank the Reviewer for this suggestion and have now expanded the discussion accordingly (lines 475-483 in the revised manuscript) and also cite the most important literature.*

Reviewer Comment #9: Reference 66 does not use CPMG profiles (line 843). The authors should carefully revise the reference.

Reply: *We thank the reviewer for pointing out the mistake and have now corrected the respective part of the main text.*